materials science

slip-spring model, constraint release, polymer chains

**Author for correspondence:**
Huifeng Tan
e-mail: tanhf@hit.edu.cn

# Slip-spring simulations of different constraint release environments for linear polymer chains

Teng Ma[1], Guochang Lin[1,2] and Huifeng Tan[1,2]

[1]Centre for Composite Materials and Structures, and [2]National Key Laboratory of Science and Technology for National Defence on Advanced Composites in Special Environments, Harbin Institute of Technology, Harbin 150080, People's Republic of China

TM, 0000-0001-7202-3728; HT, 0000-0001-7780-1410

The constraint release (CR) mechanism has important effects on polymer relaxation and the chains will show different relaxation behaviour in conditions of monodisperse, bidisperse and other topological environments. By comparing relaxation data of linear polyisoprene (PI) chains dissolved in very long matrix and monodisperse melts, Matsumiya *et al.* showed that CR mechanism accelerates both dielectric and viscoelastic relaxation (Matsumiya *et al.* 2013 *Macromolecules* **46**, 6067. (doi:10.1021/ma400606n)). In this work, the experimental data reported by Matsumiya *et al.* are reproduced using the single slip-spring (SSp) model and the CR accelerating effects on both dielectric and viscoelastic relaxation are validated by simulations. This effect on viscoelastic relaxation is more pronounced. The coincidence for end-to-end relaxation and the viscoelastic relaxation has also been checked using probe version SSp model. A variant of SSp with each entanglement assigning a characteristic lifetime is also proposed to simulate various CR environment flexibly. Using this lifetime version SSp model, the correct relaxation function can be obtained with equal numbers of entanglement destructions by CR and reptation/contour length fluctuation (CLF) for monodisperse melts. Good agreement with published experiment data is also obtained for bidisperse melts, which validates the ability to correctly describe the CR environment of the lifetime version model.

## 1. Introduction

Polymer plays an increasingly important role in industry from structural materials such as in the aerospace field to functional materials such as the polymer coating of electrodes [1–10]. Understanding of the dynamics of entangled polymer is crucial for polymer processing industry and many attempts have been made

to describe this issue in the literature. The goal in the polymer industry is to design product with perfect macroscopic properties by tailoring the chemical make-up of the polymer and the processing conditions. Molecular dynamics seems a promising tool but still not an amenable one for the moment for its enormous computational cost on the large time and length scales involved [11–14]. Tube model simplifies the dynamics by describing the movement of one single chain in a tube, formed by constraints imposed by surrounding chains [15,16]. For its success in revealing relaxation mechanism such as reptation and contour length fluctuations (CLF), tube model and its variants are used extensively in the literature [17–21]. However, constraint release (CR) mechanism owing to the motion of surrounding chains is still a challenge for tube models due to its complexity in expressing the multi-chain effects by using one single tube's movement.

CR dynamics have important effects on polymer relaxation, especially in polydisperse melts with different chain lengths. On the one hand, for the situation of very low volume fraction of long chains, fast relaxation of the short chains dissolves the constraints environment of long chains quickly, which reduce the relaxation time of long chains considerably. On the other hand, for the situation of short chains dissolved in very long chains, i.e. the so-called probe rheology in this work, constraints release of short chains is depressed due to blending and the relaxation time of short chains becomes longer. One pioneering simulation work on checking the dielectric relaxation mechanism in entangled polymers is achieved by Pilyugina *et al.* [22] They used detailed slip-link model (DSM) which is a well-defined mathematical object and has been used and extended widely in the literature [23–26]. DSM models have been valid to predict both dielectric and viscoelastic relaxations for linear monodisperse, linear bidisperse and monodisperse star-branched melts. Their results suggest no CR contribution to the end-to-end fluctuation of monodisperse and bidisperse linear polyisoprene (PI). The success of DSM model and the like inspired us to find a similar approach to separate the sliding dynamics and CR dynamics for slip-spring models. At the same time, the use of lifetime distribution probability makes it much more compute-efficient, flexible and adaptable. For its advantages, DSM and the like models can be used as a guide for other models and can be extended to nonlinear flow, polydisperse phases, coarse graining, etc. [27–29].

Matsumiya *et al.* [30] made dielectric and viscoelastic measurements for PI/PI blends with a small volume fraction of the short component (probe rheology). For the first time, they show identical mode distribution and relaxation time for the probe chain in the blend, which validate that the CR mechanism is fully suppressed for short chains. They also demonstrate the CR mechanism accelerates the dielectric relaxation and refinement should be implemented for recent modelling, which neglects the CR contribution to the dielectric relaxation.

In contrast with Matsumiya *et al.* [30], Shivokhin *et al.* [31] focus on faster CR environments and study the effect on relaxation of the end-to-end vector of long probe chains. Shivokhin *et al.* improve Likhtman's slip-spring (SSp) model [32] for clarifying the relaxation mechanisms of probe chains in various CR environments with different CR rates. Their work shows the single-chain SSp model is a good tool to investigate the viscoelastic and dielectric response of monodisperse and polydisperse systems.

Masubuchi *et al.* [33] reproduce the probe rheology data reported by Matsumiya *et al.* [30] using primitive chain network (PCN) model, which describes the dynamics of the network formed by the primitive chains and is used extensively in literature [34–38]. Their results agree with the experiment well. However, owing to possibly some flaws of orientational cross-correlation (OCC) in the model, the simulation fails to duplicate the viscoelastic relaxation intensity of the experiment. They suggest using other models with rigorous thermodynamic expression instead for obtaining consistent viscoelastic data.

The molecular weight of matrix chain is far bigger than that of probe chains in the experiment by Matsumiya *et al.* [30], thus CR mechanism was suppressed for probe chains, which is a simple CR environment and the CR effects on relaxation can be examined in the experimentally clearest way. However, in bidisperse melts or more complex system, there exist various CR environments. How to describe the CR dynamics in these systems is vital to accurate simulation of the response of the materials. CR dynamics is realized by pairing different entanglements of different chains in original Likhtman's SSp model [32], in which the corresponding slip-link (SL) will be deleted when one SL gets rid of one chain end and is destroyed. There may be some inconveniences in this implementation. First, coupling of the entanglements and the corresponding destruction/creation make the programs relatively complex. Second, it is not easy to couple entanglements on different chains in various relaxation environments. Third, for the existence of pairing entanglements, the process of computation cannot be parallelized completely. Meanwhile, as noted by Khaliullin & Schieber [39], the assumption of binary interactions of entanglement is not necessary and decoupling entanglements make studying separate relaxation contributions possible. For these reasons, one may seek some new strategies to treat more complex CR environments.

In this study, the experimental data reported by Matsumiya *et al.* [30] are reproduced using the SSp model and the conclusions are validated by simulations. The simulated viscoelastic relaxation intensity is in good agreement with the experimental data. Furthermore, we vary the SSp model with assigning every entanglement a characteristic lifetime. The lifetime distribution is determined by fitting each entanglement's lifetime from dynamics turning off the CR mechanism. It is fair to note that we have not used this model to reproduce various rheology data in the literature. We seek a strategy with identical destructions of entanglement due to reptation/CLF and CR for monodisperse melts. Bidisperse melts are also simulated and results agree with published experimental data, which validate the ability of this lifetime version model.

# 2. Model and methodology

The simulations in this work are based Likhtman's slip-spring model [32], which describes the dynamics of Rouse chain consisting of $N$ beads connected by $N-1$ springs. $Z = N/N_e$ slip-springs are randomly distributed along each chain, where $N_e$ means the average number of Kuhn segments between slip-springs. Slip-springs represent the topology constraints of surrounding chains with one end fixed at the anchoring points $a_j$ and the other end attached to one bead $r_j$ of the probe chain. The potential of the Rouse chains is $U = (3k_BT/2b^2)\sum_{i=0}^{N-1}(\mathbf{r}_{i+1} - \mathbf{r}_i)^2$ and the potential of the virtual springs is $U_{SS} = (3k_BT/2N_sb^2)\sum_{j=1}^{Z}(\mathbf{a}_j - \mathbf{r}_{S_j})^2$, in which $b$ is the monomer size and $N_s$ is the stiffness of virtual springs. The locations of the beads are updated by Brownian dynamics and the slip-springs slide or hop between polymer beads, which depends on a discrete or a continuous description. The movement of slip-spring describes the reptation-like motion of the chains. For its simplicity and easy extensibility, there are many variants of the model in literature and the detailed implementation and parameters can be found elsewhere [40–46].

We adopt the discrete version of slip-spring model as described in [31,47] and use the following steps to complete the simulations in this article:

1. Sample the model: we generate the polymer conformations by sampling from the Gaussian distribution using $P_{eq}(\{r_i\}) = (3/2\pi b^2)^{3(N-1)/2}\exp[-\sum_{i=1}^{N-1}(3/2b^2)(\mathbf{r}_{i+1} - \mathbf{r}_i)^2]$, generate Z slip-springs by uniform distribution and generate each anchoring point from the Gaussian distribution using $P_{eq}(a_j|\{r_i\}, S_j) \equiv (3/2\pi N_s b^2)^{3/2}\exp[-(3/2N_s b^2)(\mathbf{r}_{S_j} - \mathbf{a}_j)^2]$.
2. Update the conformation of polymers via Brownian dynamics using

$$\xi\mathbf{r}_i(t + \Delta t) = \left[\frac{3k_BT}{b^2}(\mathbf{r}_{i+1} - 2\mathbf{r}_i + \mathbf{r}_{i-1}) + \mathbf{f}_i(t) + \frac{3k_BT}{N_s b^2}\sum_{j x_j = i}(\mathbf{a}_j - \mathbf{r}_i)\right]\Delta t, \quad (2.1)$$

in which $\xi$ is the friction coefficient of monomer and $\mathbf{f}_i(t)$ is Gaussian white noise with zero mean, variance $\langle f_i(t)f_j(t)\rangle = 2k_BT\xi I\delta(t - t')\delta_{ij}$.
3. Update the configuration of the slip-springs via Monte Carlo moves: a discrete slip-spring jump is attempted on average per slip-spring of each chain at each time step and one bead can only occupy one single slip-spring. For this hopping rate, the friction of the slip-springs is assumed to be $\xi_{SS} = (k_BT/b^2)\tau_{SS}$ here, in which $\tau_{SS}$ is equal to the time step d$t$ here. A hopping move of the $j$th slip-spring from $r_{S_j}$ to a neighbour one $r_{S_{j\pm 1}}$ is accepted with probability $P_{accept} = \min(1, \exp(-U))$. The probability to propose the move $S_j \to S_{j+1}$ or $S_{j+1} \to S_j$ is equal to $\frac{1}{2}$. If one slip-spring reached the end of the chain, it gets rid of the chain with probability $\frac{1}{2}$. We hold constant entanglement number during the simulations, and how to recreate the new slip-spring depends on the constraint environment which will be described in detail below.

The slip-springs renewal algorithm and the description of CR mechanism are particularly important in simulating the disentanglement process of entangled polymer melts. We may adopt different strategies for describing different CR environments in this work. For simulations of monodisperse melts, when a slip-spring passes through the end of its chain, it and its corresponding slip-spring will be deleted from the ensemble. Simultaneously, one new slip-spring pair will be added to the system with one at the unoccupied chain end of randomly chosen chain and another in any place of another random chain. This renewal scheme couples the mechanism of reptation/CLF and CR and has been widely used in the literature. Here we execute discrete slip-spring jump via Metropolis Monte Carlo (MC) moves as described in [31]. We refer to this strategy as coupling entanglements version below.

**Table 1.** Basic parameters used in the SSp model.

| | | |
|---|---|---|
| thermal energy | $k_B T = 1$ | |
| monomer size | $b = 1$ | |
| friction coefficient | $\zeta = 1$ | |
| strength of slip-spring | $N_s = 0.5$ | |
| time step | $dt = 0.05$ | |

For simulations of probe rheology as the experiment by Matsumiya *et al.* [30], in which the relaxation time of matrix chain is much longer than the probe component and the CR mechanism is fully deactivated for short chains, the motion of long chain components and its effect of topological changes on short chains are ignored for computational efficiency. In the probe version program, when one slip–spring reaches the chain end and is deleted, a new slip-spring is recreated at a randomly chosen unoccupied chain end. The entanglements in the middle of the chains will not be destroyed and CR mechanism is fully suppressed (probe version) in the simulations. In the experiments of Matsumiya *et al.* [30], the dielectric relaxation and viscoelastic relaxation agree with each other indicating the CR effect totally vanishes for the short component; thus we may simulate the probe components by turning off CR dynamics completely. It is worth noting that the CLF of matrix chains is also fully suppressed, which may be not the case in the real melts.

The coupling entanglements and gel version above seem weak in dealing with more complex CR environment. Decouple the reptation/CLF and CR mechanism may make it more flexible in handling CR events. In this implementation, entanglements are not paired but with a characteristic lifetime, beyond which it will be destroyed from the system. This process mimics the entanglement on the probe chain disappearing by an imaginary corresponding chain sliding away from it, and the characteristic lifetime is the duration of this entanglement creation. With the destruction of one entanglement due to either reptation/CLF or exceeding its lifetime (CR), a new slip-spring with new assigned lifetime will be added on randomly selected chain. The position depends on the destruction type (reptation/CLF or CR) and it will be on the chain end when the slip-spring is destroyed by reptation/CLF. When all the entanglements are assigned with an infinite lifetime, entanglements will not be deleted by limited lifetime and the CR dynamics is turned off, which is the probe version model. When restricting the slip-spring can be destroyed only by lifetime manner, entanglement will not move out of the chain and thus reptation/CLF dynamics is suppressed. By decoupling of entanglements and so decoupling of reptation/CLF and CR, we can control reptation/CLF and CR mechanism separately. We refer to this strategy as lifetime version below.

The melts simulated in this paper contain PI with molecular weight of 21K, 43K, 99K, 179K and polystyrene (PS) with molecular weight of 60K and 177K. The basic simulation parameters used in this paper are listed in table 1. The number of beads $N$ of each molecular weight and corresponding numbers of entanglement $Z$ are listed in table 2. The average number of Kuhn segments between slip-springs, $N_e = N/Z \approx 4$, which is a typical value in the literature. The slight change of $N_e$ here is made to better match the experimental results. The stress relaxation function $G(t)$ is computed using the correlation functions of shear stress, including autocorrelation contributions and cross-correlation contributions from the slip-springs. The dielectric relaxation function is computed using the end-to-end vector autocorrelation functions for each chain. The expression of $G(t)$, $\varepsilon$, stress related to Rouse potential $\sigma_{\alpha\beta}^R$ and slip-spring $\sigma_{\alpha\beta}^{sL}$ are given below [42,48]:

$$G(t) = \frac{V}{k_B T} \frac{1}{3} \left\langle \sum_{\alpha=1}^{2} \sum_{\beta>\alpha}^{3} \sigma_{\alpha\beta}^R(t) \left( \sigma_{\alpha\beta}^R(0) + \sigma_{\alpha\beta}^{SL}(0) \right) \right\rangle, \tag{2.2}$$

$$\varepsilon(t) = \frac{\langle \mathbf{P}(t)\mathbf{P}(0) \rangle}{\langle P^2(0) \rangle}, \tag{2.3}$$

$$\sigma_{\alpha\beta}^R = -\frac{3k_B T}{V b^2} \left\langle \sum_{i=1}^{N} (r_{\alpha,i} - r_{\alpha,i-1})(r_{\beta,i} - r_{\beta,i-1}) \right\rangle \tag{2.4}$$

and

$$\sigma_{\alpha\beta}^{sL} = -\frac{3k_B T}{N_s V b^2} \left\langle \sum_{j=1}^{Z} (r_{\alpha,x_j} - a_{\alpha,j})(r_{\beta,x_j} - a_{\beta,j}) \right\rangle, \tag{2.5}$$

**Table 2.** Numbers of beads ($N$) and entanglements ($Z$) characterizing the melts.

| sample | $N$ | $Z$ |
|--------|-----|-----|
| PI21K | 21 | 6 |
| PI43K | 43 | 12 |
| PI99K | 99 | 28 |
| PI179K | 179 | 51 |
| PS60K | 30 | 8 |
| PS177K | 88 | 22 |

where $V$ is the volume of the melts and the indices 1, 2, 3, $\alpha$, $\beta$ represent the Cartesian coordinates. The time correlation functions are calculated on the fly using multiple-tau correlator method proposed by Ramírez *et al.* [49]. The loss and storage modulus are extracted by fitting $G(t)$ to a sum of Maxwell modes using RepTate software [50]. We obtain the relaxation modulus of polydisperse melts using

$$G(t) = \sum_{i=1}^{n} w_i G_i(t), \tag{2.6}$$

where $G_i$ is the relaxation modulus of the $i$th component and $w_i$ is associated volume fraction.

## 3. Results

Figure 1 shows the dielectric loss results at 40°C for monodisperse PI melts comparing with the experiment data. The simulated results are multiplied by $\tau = 10\,\mu s$, $G_0 = 3.3\,MPa$, $\varepsilon_0 = 0.085$, respectively. As proposed by the reviewers, the modulus should be determined using $G_0 = \rho RT/M_0$ with values $\rho = 920\,kg\,m^{-3}$, $T = 313\,K$, $R = 8.314\,m^3\,Pa\,K^{-1}\,mol^{-1}$ and $M_0 = 1\,kg\,mol^{-1}$ for PI used here. The computed $G_0 = 2.39\,MPa$, which is smaller than the fitted 3.3 MPa and here we use 3.3 MPa for better mapping to the experimental data. The simulations agree with the experiment well for identical relaxation time and intensity [30]. Figure 2 gives the viscoelastic relaxation results. It should be noted that the viscoelastic relaxation time of PI21K is overestimated though its dielectric relaxation time is quite a good match to the experiment data. This may be because of the more CR chances in experiments for short chains and reduced apparent number of entanglements. However, the simulations cannot catch the subtle change in experiment by using same $N_e$ and MC attempts for all length chains. So the simulated viscoelastic relaxation time of PI21K is longer than experimental results. In comparison, the dielectric relaxation has not been seriously affected, for the effect of CR on viscoelastic relaxation is much more significant than dielectric relaxation. At the same time, as stated by Khaliullin & Schieber [39], it is too strict to use single fitting parameter $\tau$ for all melts and one can change $\tau$ for low-molecular-weight systems slightly to achieve a good fit in these simulations.

Figure 3 shows the dielectric loss of probe chains in comparison to the monodisperse melts and the associated experimental data. Figure 4 shows the viscoelastic loss data. The relaxation of the probe chains is slower than in the monodisperse melts, which agrees with the conclusion that CR mechanism accelerates both dielectric relaxation and viscoelastic relaxation of monodisperse melts by Matsumiya *et al.* [30]. From figures 3 and 4, the retardation of viscoelastic relaxation of the probe chains is more significant than the dielectric relaxation and the viscoelastic mode distribution changes greater due to turning of CR mechanism in comparison to dielectric mode distribution. These results all agree well with experiments [30].

For better fit of monodisperse results, the timescale of simulations was chosen as $\tau = 10\,s$. A negative influence of that is the simulated dielectric relaxation time of probe chains is underestimated systematically. We may chose a different MC strategy and tune the timescale to improve the discrepancy. However, the MC strategy in this paper does not change the conclusions. The dielectric relaxation time of probe chains and monodisperse chains have almost no difference for PI179K. With increasing the molecular weight of the chains, the CR effect becomes small gradually and we can infer this effect of CR can be neglected when the molecular weight of chains is large enough.

For the inaccurate viscoelastic relaxation time for 21K, we check the relaxation time and intensity of viscoelastic relaxation for 43K. We note that the relaxation time and intensity of viscoelastic relaxation for

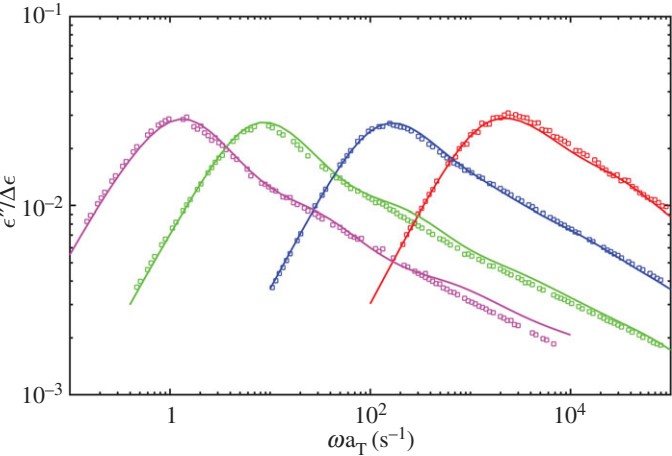

**Figure 1.** Dielectric loss results at 40°C for monodisperse PI melts of 179K (magenta), 99K (green), 43K (blue) and 21K (red). Symbols are experiment data from Matsumiya *et al.* [30].

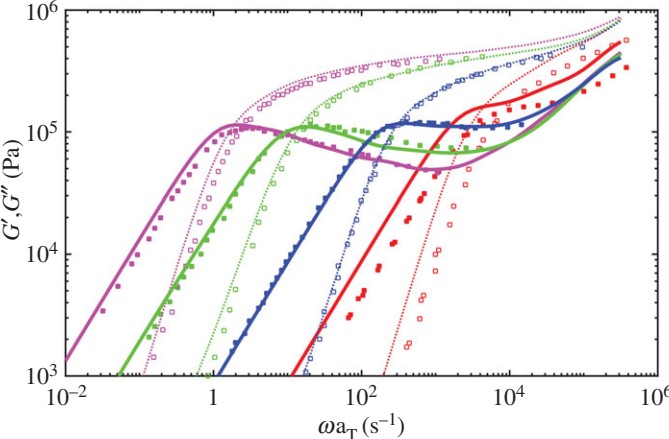

**Figure 2.** Viscoelastic storage (dotted curves) and loss modulus (solid curves) from the simulations at 40°C for monodisperse PI melts of 179K (magenta), 99K (green), 43K (blue) and 21K (red). Symbols are experiment data from Matsumiya *et al.* [30].

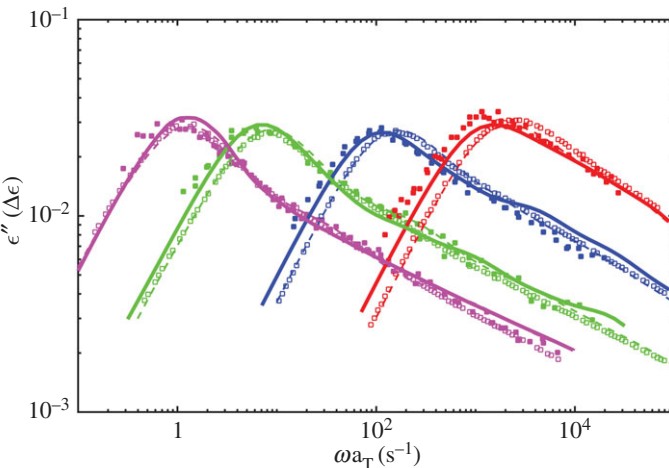

**Figure 3.** Simulated dielectric loss results of probe chains in comparison to the monodisperse melts with molecular weights of 179K (magenta), 99K (green), 43K (blue) and 21K (red). Solid lines and broken ones represent the simulation results for probe chains and monodisperse melts, respectively. Symbols are associated experiment data reproduced from Matsumiya *et al.* [30].

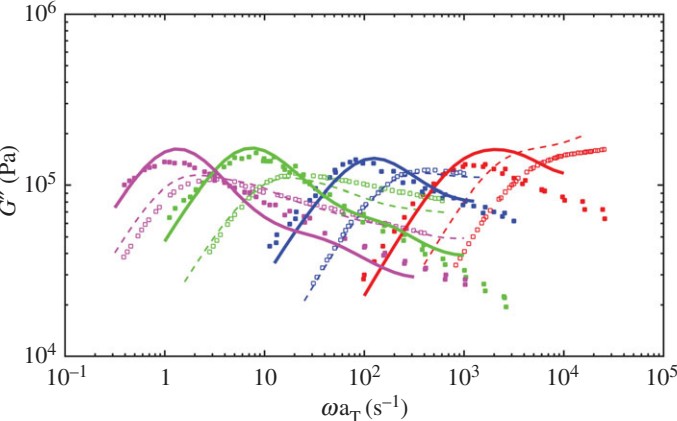

**Figure 4.** Simulated viscoelastic loss results of probe chains in comparison to the monodisperse melts with molecular weights of 179K (magenta), 99K (green), 43K (blue) and 21K (red). Solid lines and broken ones represent the simulation results for probe chains and monodisperse melts, respectively. Symbols are associated experiment data reproduced from Matsumiya *et al.* [30].

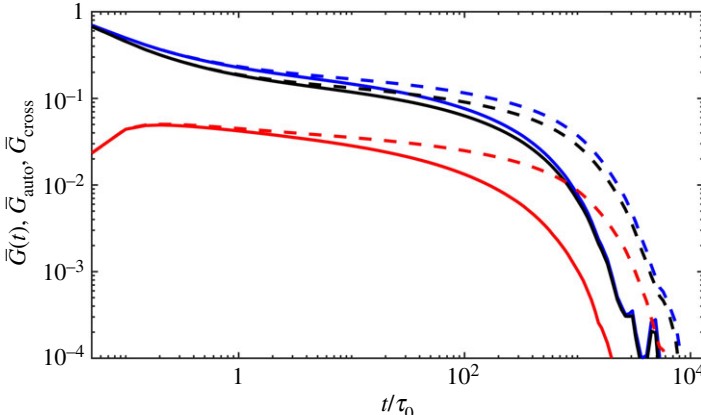

**Figure 5.** Simulated stress relaxation functions (blue), the autocorrelation contribution (black) and the cross-correlation contribution (red) of probe chains (broken lines) and the monodisperse melts (solid lines) with molecular weights of 43K.

43K obtained through PCN [33] are not consistent with the experiment. In the experiment by Matsumiya *et al.* [30] and simulations in this work, the probed viscoelastic loss is larger than that of the monodisperse melts around the peak whereas PCN simulations show a contrary tendency. Meanwhile, the intensity is molecular weight-dependent from PCN, which is wrong and the SSp results recover this irrelevance found in the experiment. The viscoelastic relaxation function with the autocorrelation and cross-correlation contributions are shown in figure 5 for both probe rheology and monodisperse melts of PI43K. As noted by the authors, the failure of PCN in predicting the viscoelastic intensity of probe chains is because of the flaws for OCC in the PCN model. Here we only show the results of SSp can reproduce the experimental data and it should not be confused with the OCC in the PCN model. The OCC in the PCN model is a multi-chain property and results from force balance at entanglement points.

To check the coincidence for the end-to-end relaxation and the segment relaxation under probe conditions, the dielectric and viscoelastic relaxations of the probe chains are compared. The dielectric loss data is multiplied by a shift-factor A in figure 6 for comparison purpose. The relaxations agree with each other for all four melts. The deviation of mode distribution in the high-frequency region should not be confused with that of the experimental data in which the long component relaxation plays a part, whereas the CR mechanism in this simulation is fully suppressed.

In this section the agreements between the SSp model predicted and the experiment data support that the SSp model is suitable for both monodisperse melt and probe CR environment. The conclusions made by Matsumiya *et al.* [30] have been validated using simulations. The CR accelerating effect and the coincidence of dielectric and viscoelastic relaxation have been verified using SSp model. Though the probe SSp simulations can obtain correct conclusions, there may be some shortages. First, it

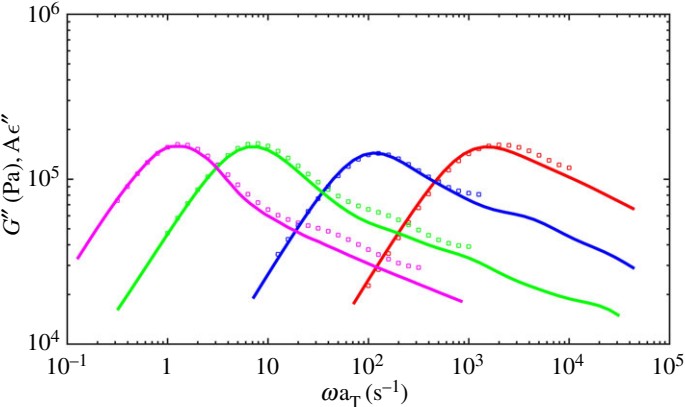

**Figure 6.** Comparision of simulated viscoelastic loss (solid lines) and dielectric loss (unfilled symbols) of probe chains with molecular weights of 179K (magenta), 99K (green), 43K (blue) and 21K (red). The dielectric loss data is multiplied by a shift-factor A to match its peak height with that of the viscoelastic loss data.

should be noted that the probe version simulations cannot reflect the difference of different volume fractions. Second, the CLF of matrix chains has been totally suppressed which would affect the results. Meanwhile, the computational efficiency is low for long chains and further improvements should be made to realize a more flexible and efficient SSp model.

# 4. The lifetime distribution and application of lifetime version SSp

Next we seek strategy to achieve the lifetime version SSp mentioned in §2. The lifetime of entanglements is related to the dynamics of polymer chains and one entanglement disappears by either sliding away from the end of the chain which it is entangled with or vice versa. The former is reptation/CLF and the latter is constraint release (CR). In the original single-chain slip-spring model when one slip-spring slips away from the chain end, there should be one corresponding slip-spring disappearing. For the lifetime version SSp, the lifetime distribution of entanglements should be obtained and the entanglements destroyed by reptation/CLF should be compared with CR. This section gives the lifetime distribution based on [39]. Khaliullin *et al.* [39,51] model CR self-consistently using the DSM model, and their numbers of entanglements deleted by sliding dynamics (SD) and constraint dynamics (CD) are equal. The cumulative distribution of survival time of entanglements upon their individual birth time is equal to $1 - f_d(t)$, in which $f_d(t)$ is the time derivative of the fraction of survived entanglements $f(t)$. The concrete expression of $f(t)$ is $f(t) = \int_0^\infty p^{CR} \exp\left(-(t/\tau^{CR})\right) d\tau^{CR}$, which means the survival time of entanglements from any moment. We first get $f_d(t)$ curve by simulations without CR mechanism using the probe version model (or gel version). Here we use results of sample of PI21K because it is the most time-saving of all and other samples show similar results. We first simulate 100 PI21K chains for $10^5$ time steps using the probe version SSp. During the dynamics, each entanglement's lifetime from its birth to its disentanglement was recorded and the normalized curve $f_d(t)$ was obtained. Using the curve we infer the probability form of lifetime distribution and then fit the curve using one expression discussed below by nonlinear least-squares method. The parameters in the probability expression will be eventually obtained. Figures 7 and 8 show the $f_d(t)$ curve on log–log scale and on semilog scale, respectively, which shows the relaxation spectrum also has curved power-law region and a single exponential region as shown by the DSM model [39] results. The power-law region is considered a result of fast CLF motion and the single exponential term corresponds to reptation. Therefore, the form of normalized lifetime distribution probability is assumed as

$$p^{CR}(\tau) = \frac{(1-g)\alpha}{\tau_{max}^\alpha - \tau_0^\alpha} \tau^{\alpha-1} H(\tau - \tau_0) H(\tau_{max} - \tau) + g\delta(\tau_d - \tau), \tag{4.1}$$

which is eqn (21) in [39]. Then we look for suitable $g, \alpha, \tau_0, \tau_{max}, \tau_d$ in equation (4.1) to satisfy two conditions: get the stress relaxation function which agrees with coupling entanglements version results well using these parameters and delete an equal number of entanglements by reptation/CLF and CR, which make the model

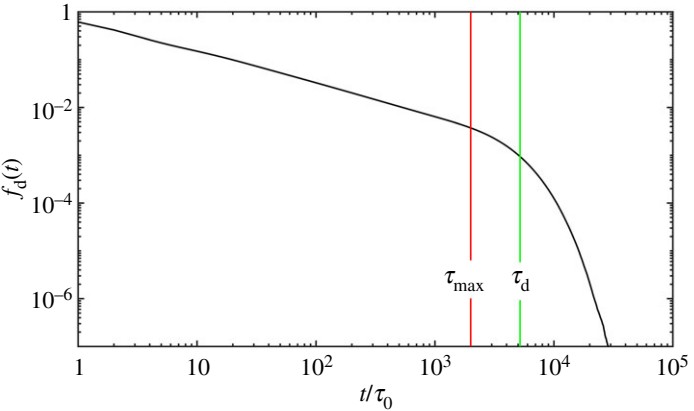

**Figure 7.** $f_d(t)$ curve on log–log scale from 100 chains simulation for PI21K, which shows the power-law region. The fitted $\tau_{max}$ and $\tau_d$ are also shown. $\tau_0$ is not shown because it smaller than d$t$.

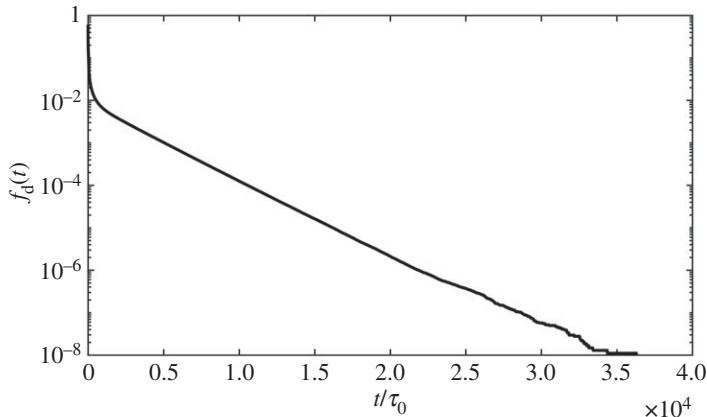

**Figure 8.** $f_d(t)$ curve on semilog scale from 100 chains simulation for PI21K, which shows the single exponential region.

self-consistent. Similar to eqn (18) in [39], we fit the $f_d(t)$ curve using new form below:

$$
f_d(t) = \frac{\int_0^\infty \left[ \frac{(1-g)\alpha}{\tau_{max}^\alpha - \tau_0^\alpha} \tau^{\alpha-1} H(\tau - \tau_0) H(\tau_{max} - \tau) H(\tau - t) + \frac{g}{\tau} \delta(\tau_d - \tau) e^{-t/\tau} \right] d\tau}{\int_0^\infty \left[ \frac{(1-g)\alpha}{\tau_{max}^\alpha - \tau_0^\alpha} \tau^{\alpha-1} H(\tau - \tau_0) H(\tau_{max} - \tau) + \frac{g}{\tau} \delta(\tau_d - \tau) \right] d\tau}. \tag{4.2}
$$

As noted by Khaliullin *et al.* [39], the parameters can be obtained by iteration and here we use the first fitting results for monodisperse melts. Due to blending, the relaxation environment has changed greatly and we iterate the procedure for bidisperse melts. We restrict that CR creates entanglements using only the power-law parts of equation (4.1), which creates short-lived entanglements more frequently than long-lived ones due to more frequent destructions of short-lived entanglements. By the least-squares method of relative residuals, we obtain the parameters (listed in table 3) and find simulations using them satisfy the two conditions mentioned above well. The results are shown in figures 9 and 10. We also obtain the parameters by fitting eqn (19) in [39]. The fitted parameters are listed in table 3 and the relaxation function and deleted entanglement numbers are depicted in figures 9 and 10. In comparison to coupling entanglements version results, the relaxation is slow and destruction by CR is underestimated. The difference may be from the different description of dynamics between the SSp model and the DSM model. The comparison between these two models may be an interesting topic in the future research.

Shivokhin *et al.* [31] uses exponential form distribution to express different CR environment for studying the effect of different relaxation times on relaxation. We also show the results using exponential distribution with mean equal to the $\tau_d$ for comparison in figures 9 and 10. It can be found that the times of disentanglement by CR is far less than that by reptation/CLF. For short chains, the

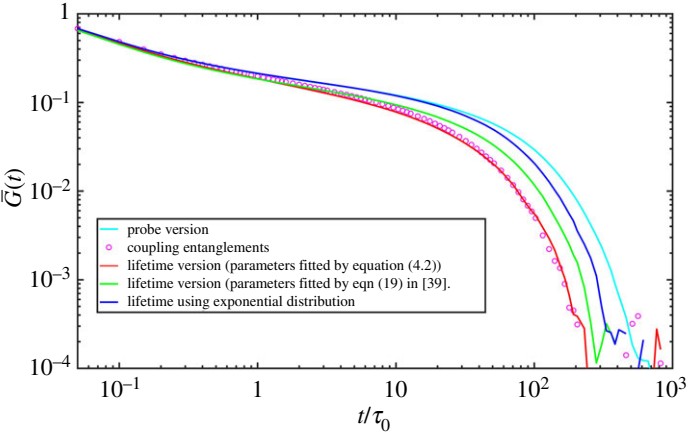

**Figure 9.** Simulated relaxation functions for PI21K of different versions: probe version, coupling entanglements version, lifetime version using parameters fitted by equation (4.2) in this work and eqn (19) in [39], respectively. The results using exponential distribution are also shown.

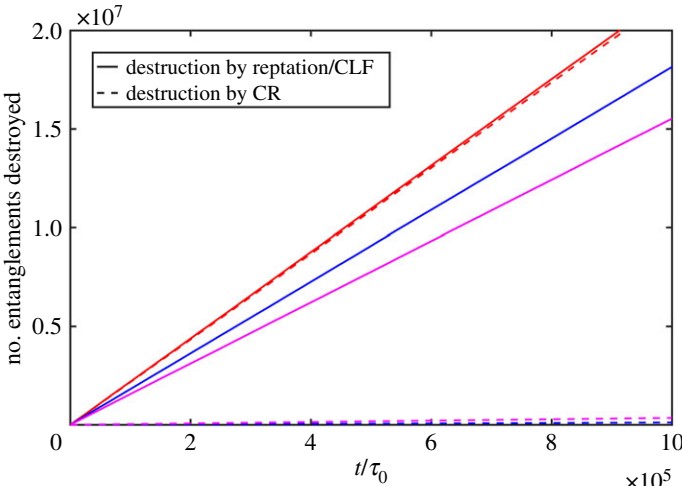

**Figure 10.** Simulated numbers of entanglements destroyed for PI21K with 100 chains. The solid lines represent destruction by reptation/CLF and broken lines represent destruction by CR. The red and magenta curves represent results using parameters fitted by equation (4.2) in this work and eqn (19) in [39], respectively. The blue curves represent results using lifetime from exponential distribution.

**Table 3.** Fitted parameters used in lifetime version SSp for PI21K.

| fitting forms or distribution form | $g$ | $\alpha$ | $\tau_0/\mathrm{d}t$ | $\tau_{max}/\mathrm{d}t$ | $\tau_d/\mathrm{d}t$ |
|---|---|---|---|---|---|
| equation (4.2) in this work | 0.67 | −0.607 | 0.45 | 2000 | 5200 |
| equation (19) in [39] | 0.67 | 0.35 | 0.63 | 2000 | 5200 |
| exponential distribution | — | — | — | — | 5200 |

results using the exponential form distribution in which only the reptation mechanism is considered show big discrepancy and it may be better for longer chains with more entanglements.

To check the reptation/CLF and CR mechanism of the lifetime version model, we model bidisperse melts comprising long and short chains, in which the CR environment is more complex. For the existing data in literature and computational efficiency, we consider bidisperse melts comprising PS177 and PS60. The volume fraction of long component is 0.4 and 0.6 respectively. The experiment data of PS177&60

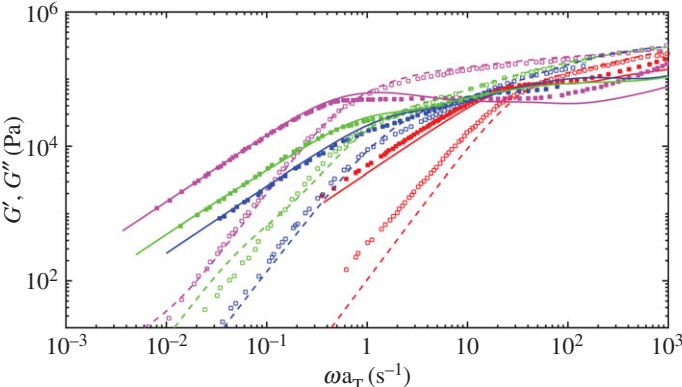

**Figure 11.** Simulated viscoelastic storage (broken lines) and loss modulus (solid lines) for PS60 (red), PS177 (magenta), PS60&177% 40 (blue) and PS60&177%60 (green). The symbols are experiment data reproduced from [52].

**Table 4.** Fitted parameters used in lifetime version SSp for PS177&60.

| parameters | PS60 (mono) | PS177 (mono) | PS60 (blend) | PS177 (blend) |
|---|---|---|---|---|
| $g$ | 0.76 | 0.91 | 0.65 | 0.849 |
| $\alpha$ | −0.603 | −0.599 | −0.602 | −0.601 |
| $\tau_0$ | 0.45 | 0.45 | 0.45 | 0.41 |
| $\tau_{max}$ | 6000 | 40 000 | 4800 | 5300 |
| $\tau_d$ | 10 500 | 80 000 | 14 800 | 46 000 |

blends can be found in [52]. The detail parameters such as beads number of each chain $N_i$ and entanglements number $Z_i$ are listed in table 2 and we use $i = 1$ and 2 for the long and short components. The numbers of chains of each component in the blend are determined by the volume fractions. The fitted parameters of lifetime distribution used in lifetime version SSp for PS177&60 are obtained by PS60&177%60 simulations iterated manually and are listed in table 4. For the cost of computation, we also use the parameters for PS60&177%40. For bidisperse melts, when one slip-spring of certain type (e.g. short–long entanglements) reaches its lifetime limit, a new slip-spring of the same type will be added into the system and assigned a random lifetime from relevant distributions. Figure 11 gives the simulation results of both monodisperse and bidisperse melts and the results from simulations show agreement with experimental measurements from [52].

The lifetime version SSp inherits advantages of SSp such as simplicity and easy extension, at the same time as it discards the coupling of entanglements but uses characteristic lifetime instead. This change makes it more flexible to simulate complex CR environment and more easily controlled to validate some theories of CR. It should be noted that the lifetime version model used in this section is still a rough model. Further refinement is necessary and the effects of the parameters of the lifetime distributions should be well understand to achieve more complex CR environment.

## 5. Conclusion

The results reported by Matsumiya *et al.* [30] have been reproduced by SSp model and some conclusions have been validated by simulations. CR mechanism accelerates both dielectric and viscoelastic relaxation in monodisperse melts. The coincidence of dielectric and viscoelastic relaxation was captured when CR mechanism was fully suppressed. The simulated viscoelastic relaxation intensity is in good agreement with the experimental data of both probe rheology and monodisperse melts. However, the CLF of long chains has been ignored in SSp simulations, which may affect the results. For more complex CR environment, a lifetime version SSp has been presented and the CR parameters have been determined self-consistently. Using lifetime version SSp, monodisperse and bidisperse melts relaxation have been simulated and the results agree well with experiment data in the literature.

Ethics. This article does not present research with ethical considerations.

Data accessibility. Data available from the Dryad Digital Repository: http://dx.doi.org/10.5061/dryad.b14bh40 [53].

Authors' contributions. H.T. and G.L. participated in the study design. T.M. carried out the simulations and the data analysis. T.M. drafted the initial manuscript. All authors contributed to the interpretation of the results and manuscript revisions. All authors gave final approval for publication.

Competing interests. We declare we have no competing interests.

Funding. This project was supported by the Foundation for Innovative Research Groups of the National Natural Science Foundation of China (grant no. 11421091) and the Fundamental Research Funds for the Central Universities (grant no. HIT. MKSTISP.2016 09).

Acknowledgements. The authors thank the reviewers for reviewing the manuscript.

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
