## [Reviewer comments · Royal Society Open Science]

Review History

RSOS-191046.R0 (Original submission)

Review form: Reviewer 1

Is the manuscript scientifically sound in its present form?

Yes

Are the interpretations and conclusions justified by the results?

Yes

Is the language acceptable?

No

Do you have any ethical concerns with this paper?

No

Have you any concerns about statistical analyses in this paper?

No

Recommendation?

Accept with minor revision (please list in comments)

Comments to the Author(s)

The manuscript presents computer simulation studies of the dielectric loss and linear viscoelasticity of polymer melts from coarse-grained models. The authors demonstrate that the model parameters of the slip-spring model can be chosen to obtain reasonably good fits to literature data of experimental results on monodisperse, linear polyisoprene. The more original contribution made here is the investigation of a variant of the slip-spring model, where pairing of constraints is replaced by a lifetime probability, very similar to corresponding slip-link model by Schieber et al. The authors show that this variant of the model can be fitted to reasonably reproduce experimental results on mono- and bi-disperse melts. I consider the lifetime-version of the slip-spring model an interesting approach that could inspire future work due to its flexibility.

I have just few technical comments:

Table 3: I guess these parameters correspond to 21K in Fig. 9, but the authors should make this clear in the manuscript. Furthermore, "eqn (3)" does not exist, probably (4.1) is meant?

Fig 7, 8: How did the authors determine the lifetime distribution? How are they able to numerically calculate this quantity with an accuracy better than 10^{-7} ?

page 11: "simulations are in quantitative agreement ..." seems an overstatement, since the loss plateau for 60% (green) in Fig. 11 is about a factor of 2 off.

Table 4: Is the exponent alpha really that sensitive that -0.603 and -0.599 give different results from -0.6?

A number of typos and awkward wordings should be corrected, such as "mels" -> melts, "simpleness" -> simplicity, etc.

Review form: Reviewer 2

Is the manuscript scientifically sound in its present form?

Yes

Are the interpretations and conclusions justified by the results?

Yes

Is the language acceptable?

No

Do you have any ethical concerns with this paper?

No

Have you any concerns about statistical analyses in this paper?

No

Recommendation?

Reject

Comments to the Author(s)

The question I am trying to answer in my mind is "is this simply derivative work?" I note from the referee guidelines it says: "Submissions should sufficiently advance scientific knowledge. Negative findings, meta-analyses and studies testing reproducibility of significant work are also encouraged. Repeated experiments will only be considered if they provide a meaningful

contribution to the literature. Derivative work will not be considered." I am concerned that this paper might fall into the last category.

Here are the facts:

- the authors have recreated a previously published simulation algorithm (the "slip-spring model", a type of slip-link model)
- they have used it to match some previously published experimental data. So far as I know, the algorithm has not been used for specifically these data before, but it has been used on fairly similar data.
- they have then modified the slip-spring algorithm to include an idea that other authors (ref [31]) used on a slightly different slip-link model, the discrete slip-link model. But the idea is essentially the same and the two models are not so different (both slip-link models).

I admit this is a marginal judgement. There are two (fairly minor) things in the paper which arguably have not been done before: using an algorithm on a set of data it wasn't used for before, and incorporating the idea of ref [31] into specifically this set of data. but I feel the work is still quite derivative in nature, it does not break new ground.

Here are some more detailed comments about further aspects of the paper.

1) On the topic of orientational cross correlations (OCC) I feel the authors have a misunderstanding of the issue, throughout the paper. OCC in the (multi-chain) Masubuchi work, and in the single chain "slip spring" simulations are quite different things, and cannot be compared one against the other. In the Masubuchi work, the OCC is between different stress-carrying chains in the simulation and arise as a result of force balance at entanglement points between those chains. Since those simulation chains are supposed to represent real chains in a polymer melt, the question is whether real chains also have the orientational cross correlations, and that is a meaningful question (but not one that is answered by the work of the present paper).

On the other hand, the present simulations are all "single chain" simulations in that one chain does not directly exchange forces with others - there can be no OCC between different chains. The OCC are between chains and the virtual springs which hold the chains. The virtual springs are not considered to "really" carry stress. However, the OCC must be included in the Green-Kubo formula to calculate $G(t)$, even though it is considered that the real stress is only carried by the chains. So, this is an entirely different situation from the Masubuchi simulations, and not comparable at all.

2) The list of parameters in Table 1 should include Ne_{ss} .

3) The version of Reptate referred to is no longer available, and the web address no longer works. I believe there is a new Reptate available.

4) Page 6: The authors write "It's fair to note that the simulated dielectric relaxation time of probe chains is underestimated systematically". Potentially the problem is the other way around... the authors should fix the timescale of simulations using the "simpler" probe data (where CR is suppressed) and then model the more complicated CR environment of pure melt. Perhaps the pure melt predictions are actually the things that are wrong?

5) Page 8, the authors write, "The concrete expression of $f(t)$ can be found in ref. 31.". I think the present paper should be self contained and all necessary equations written here.

6) Page 9 onwards discusses the exponential form for CR events used by Shivokhin. It is worth noting that the exponential distribution was used as a test "theoretical" case to explore the effects of a single dominant CR timescale, without necessarily expecting it to be an accurate model for real chains. Further more (figure 9) the exponential distribution is bound to be worst for the

shortest chains with fewest entanglements, since such chains have a greater effect of contour length fluctuation. Longer chains should show less discrepancy.

7) For the binary blend case (figure 11) it would be strongly advisable to iterate the procedure for finding $f(t)$, not just take the first iteration. The presence of short chains can greatly accelerate the relaxation of longer chains, so the entanglement lifetime distribution is quite different from the pure melts of the two components in some (but not all) cases.

8) The magenta line in figure 11 shows a spurious relaxation near $\omega = 10\text{s}^{-1}$. What is the source of this?

9) Page 11, the following two sentences need to be either removed or elaborated on (how would one do this?)

" For example, it may be extended to study the partially dilated tube diameters to check partially DTD."

"Further refinement is necessary and the effects of the parameters of the lifetime distributions should be well understand to achieve more complex CR environment such as nanocomposites with rod-like filler"

(neither seem obviously do-able to me)

Decision letter (RSOS-191046.R0)

29-Jul-2019

Dear Dr Ma,

The editors assigned to your paper ("Slip-spring simulations of different constraint release environments for linear polymer chains") have now received comments from reviewers. We would like you to revise your paper in accordance with the referee and Associate Editor suggestions which can be found below (not including confidential reports to the Editor). Please note this decision does not guarantee eventual acceptance.

Please submit a copy of your revised paper before 21-Aug-2019. Please note that the revision deadline will expire at 00.00am on this date. If we do not hear from you within this time then it will be assumed that the paper has been withdrawn. In exceptional circumstances, extensions may be possible if agreed with the Editorial Office in advance. We do not allow multiple rounds of revision so we urge you to make every effort to fully address all of the comments at this stage. If deemed necessary by the Editors, your manuscript will be sent back to one or more of the original reviewers for assessment. If the original reviewers are not available, we may invite new reviewers.

When submitting your revised manuscript, you must respond to the comments made by the referees and upload a file "Response to Referees" in "Section 6 - File Upload". Please use this to document how you have responded to the comments, and the adjustments you have made. In

order to expedite the processing of the revised manuscript, please be as specific as possible in your response.

- Data accessibility

If you wish to submit your supporting data or code to Dryad (<http://datadryad.org/>), or modify your current submission to dryad, please use the following link:
<http://datadryad.org/submit?journalID=RSOS&manu=RSOS-191046>

- Competing interests

- Authors' contributions

- Acknowledgements

- Funding statement

Once again, thank you for submitting your manuscript to Royal Society Open Science and I look

forward to receiving your revision. If you have any questions at all, please do not hesitate to get in touch.

on behalf of Professor Hazel Assender (Associate Editor) and R. Kerry Rowe (Subject Editor)
 openscience@royalsociety.org

Associate Editor's comments (Professor Hazel Assender):

One of the reviewers has recommend rejection of this manuscript. If you wish to submit an amended version of the manuscript, it would require major revision, addressing each of the points raised by the reviewers, but also specifically outlining the novelty of the work, how it is not simply derivative of the previous work you cite, and the significance of the contribution of the work.

Comments to Author:

Reviewers' Comments to Author:
 Reviewer: 1

Comments to the Author(s)

The manuscript presents computer simulation studies of the dielectric loss and linear viscoelasticity of polymer melts from coarse-grained models. The authors demonstrate that the model parameters of the slip-spring model can be chosen to obtain reasonably good fits to literature data of experimental results on monodisperse, linear polyisoprene. The more original contribution made here is the investigation of a variant of the slip-spring model, where pairing of constraints is replaced by a lifetime probability, very similar to corresponding slip-link model by Schieber et al. The authors show that this variant of the model can be fitted to reasonably reproduce experimental results on mono- and bi-disperse melts. I consider the lifetime-version of the slip-spring model an interesting approach that could inspire future work due to its flexibility.

I have just few technical comments:

Table 3: I guess these parameters correspond to 21K in Fig. 9, but the authors should make this clear in the manuscript. Furthermore, "eqn (3)" does not exist, probably (4.1) is meant?

Fig 7, 8: How did the authors determine the lifetime distribution? How are they able to numerically calculate this quantity with an accuracy better than 10^{-7} ?

page 11: "simulations are in quantitative agreement ..." seems an overstatement, since the loss plateau for 60% (green) in Fig. 11 is about a factor of 2 off.

Table 4: Is the exponent alpha really that sensitive that -0.603 and -0.599 give different results from -0.6?

A number of typos and awkward wordings should be corrected, such as "mels" -> melts, "simpleness" -> simplicity, etc.

Reviewer: 2

Comments to the Author(s)

The question I am trying to answer in my mind is "is this simply derivative work?" I note from the referee guidelines it says: "Submissions should sufficiently advance scientific knowledge. Negative findings, meta-analyses and studies testing reproducibility of significant work are also encouraged. Repeated experiments will only be considered if they provide a meaningful contribution to the literature. Derivative work will not be considered." I am concerned that this paper might fall into the last category.

Here are the facts:

- the authors have recreated a previously published simulation algorithm (the "slip-spring model", a type of slip-link model)
- they have used it to match some previously published experimental data. So far as I know, the algorithm has not been used for specifically these data before, but it has been used on fairly similar data.
- they have then modified the slip-spring algorithm to include an idea that other authors (ref [31]) used on a slightly different slip-link model, the discrete slip-link model. But the idea is essentially the same and the two models are not so different (both slip-link models).

I admit this is a marginal judgement. There are two (fairly minor) things in the paper which arguably have not been done before: using an algorithm on a set of data it wasn't used for before, and incorporating the idea of ref [31] into specifically this set of data. but I feel the work is still quite derivative in nature, it does not break new ground.

Here are some more detailed comments about further aspects of the paper.

1) On the topic of orientational cross correlations (OCC) I feel the authors have a misunderstanding of the issue, throughout the paper. OCC in the (multi-chain) Masubuchi work, and in the single chain "slip spring" simulations are quite different things, and cannot be compared one against the other. In the Masubuchi work, the OCC is between different stress-carrying chains in the simulation and arise as a result of force balance at entanglement points between those chains. Since those simulation chains are supposed to represent real chains in a polymer melt, the question is whether real chains also have the orientational cross correlations, and that is a meaningful question (but not one that is answered by the work of the present paper).

On the other hand, the present simulations are all "single chain" simulations in that one chain does not directly exchange forces with others - there can be no OCC between different chains. The OCC are between chains and the virtual springs which hold the chains. The virtual springs are not considered to "really" carry stress. However, the OCC must be included in the Green-Kubo formula to calculate $G(t)$, even though it is considered that the real stress is only carried by the chains. So, this is an entirely different situation from the Masubuchi simulations, and not comparable at all.

2) The list of parameters in Table 1 should include Ne_{ss} .

3) The version of Reptate referred to is no longer available, and the web address no longer works. I believe there is a new Reptate available.

4) Page 6: The authors write "It's fair to note that the simulated dielectric relaxation time of probe chains is underestimated systematically". Potentially the problem is the other way around... the authors should fix the timescale of simulations using the "simpler" probe data (where CR is suppressed) and then model the more complicated CR environment of pure melt. Perhaps the pure melt predictions are actually the things that are wrong?

5) Page 8, the authors write, "The concrete expression of $f(t)$ can be found in ref. 31.". I think the present paper should be self contained and all necessary equations written here.

6) Page 9 onwards discusses the exponential form for CR events used by Shivokhin. It is worth noting that the exponential distribution was used as a test "theoretical" case to explore the effects of a single dominant CR timescale, without necessarily expecting it to be an accurate model for real chains. Further more (figure 9) the exponential distribution is bound to be worst for the shortest chains with fewest entanglements, since such chains have a greater effect of contour length fluctuation. Longer chains should show less discrepancy.

7) For the binary blend case (figure 11) it would be strongly advisable to iterate the procedure for finding $f(t)$, not just take the first iteration. The presence of short chains can greatly accelerate the relaxation of longer chains, so the entanglement lifetime distribution is quite different from the pure melts of the two components in some (but not all) cases.

8) The magenta line in figure 11 shows a spurious relaxation near $\omega = 10s^{-1}$. What is the source of this?

9) Page 11, the following two sentences need to be either removed or elaborated on (how would one do this?)

" For example, it may be extended to study the partially dilated tube diameters to check partially DTD."

"Further refinement is necessary and the effects of the parameters of the lifetime distributions should be well understand to achieve more complex CR environment such as nanocomposites with rod-like filler"

(neither seem obviously do-able to me)

Author's Response to Decision Letter for (RSOS-191046.R0)

See Appendix A.

RSOS-191046.R1 (Revision)

Review form: Reviewer 1

Is the manuscript scientifically sound in its present form?

Yes

Are the interpretations and conclusions justified by the results?

Yes

Is the language acceptable?

Yes

Do you have any ethical concerns with this paper?

No

Have you any concerns about statistical analyses in this paper?

No

Recommendation?

Accept as is

Comments to the Author(s)

I am happy with the authors response.

Review form: Reviewer 3

Is the manuscript scientifically sound in its present form?

Yes

Are the interpretations and conclusions justified by the results?

Yes

Is the language acceptable?

Yes

Do you have any ethical concerns with this paper?

No

Have you any concerns about statistical analyses in this paper?

No

Recommendation?

Major revision is needed (please make suggestions in comments)

Comments to the Author(s)

The results look as expected, so they probably calculated everything correctly. However, there are not sufficient details given. Also, they have not cited our prior work in the area, but appear to have done significantly less. The idea of modifying the slip-spring simulations in this way is actually taken over from our ongoing work, and it seems that proper credit is missing. Moreover, we have looked at this question before, using our slip link model [Pilyugina, E.; Schieber, J. D. & Andreev, M. Dielectric relaxation as an independent examination of relaxation mechanisms in entangled polymers using the discrete slip-link model *Macromolecules*, 2012, 45, 5728-5743]. There are other papers from our work that are relevant and should also be cited. We have analytic expression for $f_d(t)$ which might be compared to those found here, for example. We also have a more fine-grained version of our model that should be very similar.

In principle, I like that they are working to make the slip-spring simulations more rigorous, but am disappointed that they are largely ignoring our contributions that have largely solved all of these problems. Have they found any new conclusions?

Finally, I have a number of important technical issues.

1) Why are the authors treating the modulus as an adjustable parameter? It should be determined completely from other known quantities, like temperature, density and N_e .

2) How many beads were used per slip-spring? How was that value chosen or determined?

- 3) there should be at least two frictions in the model: between the bead and background, and a friction associated with the hopping rate of the slip-spring. That latter does not appear to be specified anywhere. How do the results depend on these values? How were they chosen?
- 4) Do the slip-springs hop or slide? The latter is problematic from a thermodynamic point of view.
- 5) How do they calculate stress? Do the virtual springs contribute? If so, then do they violate stress-optic rule? If not, then the model presumably violates thermodynamics.
- 6) How do they calculate the dielectric relaxation?

Decision letter (RSOS-191046.R1)

13-Nov-2019

Dear Dr Ma:

Manuscript ID RSOS-191046.R1 entitled "Slip-spring simulations of different constraint release environments for linear polymer chains" which you submitted to Royal Society Open Science, has been reviewed. The comments of the reviewer(s) are included at the bottom of this letter.

Please note that multiple rounds of revision are not generally permitted and no further rounds will be acceptable if your manuscript is not ready for publication following this revision.

Please submit a copy of your revised paper before 06-Dec-2019. Please note that the revision deadline will expire at 00.00am on this date. If we do not hear from you within this time then it will be assumed that the paper has been withdrawn. In exceptional circumstances, extensions may be possible if agreed with the Editorial Office in advance. We do not allow multiple rounds of revision so we urge you to make every effort to fully address all of the comments at this stage. If deemed necessary by the Editors, your manuscript will be sent back to one or more of the original reviewers for assessment. If the original reviewers are not available we may invite new reviewers.

- Ethics statement

If your study uses humans or animals please include details of the ethical approval received, including the name of the committee that granted approval. For human studies please also detail

whether informed consent was obtained. For field studies on animals please include details of all permissions, licences and/or approvals granted to carry out the fieldwork.

- Data accessibility

- Competing interests

- Authors' contributions

- Acknowledgements

- Funding statement

Kind regards,

Andrew Dunn

on behalf of Professor Hazel Assender (Associate Editor) and R. Kerry Rowe (Subject Editor)
openscience@royalsociety.org

Associate Editor Comments to Author (Professor Hazel Assender):

The issues raised by reviewer 2 should be addressed.

Reviewer comments to Author:

Reviewer: 1

Comments to the Author(s)

I am happy with the authors response.

Reviewer: 3

Comments to the Author(s)

The results look as expected, so they probably calculated everything correctly. However, there are not sufficient details given. Also, they have not cited our prior work in the area, but appear to have done significantly less. The idea of modifying the slip-spring simulations in this way is actually taken over from our ongoing work, and it seems that proper credit is missing. Moreover, we have looked at this question before, using our slip link model [Pilyugina, E.; Schieber, J. D. & Andreev, M. Dielectric relaxation as an independent examination of relaxation mechanisms in entangled polymers using the discrete slip-link model *Macromolecules*, 2012, 45, 5728-5743]. There are other papers from our work that are relevant and should also be cited. We have analytic expression for $f_d(t)$ which might be compared to those found here, for example. We also have a more fine-grained version of our model that should be very similar.

In principle, I like that they are working to make the slip-spring simulations more rigorous, but am disappointed that they are largely ignoring our contributions that have largely solved all of these problems. Have they found any new conclusions?

Finally, I have a number of important technical issues.

- 1) Why are the authors treating the modulus as an adjustable parameter? It should be determined completely from other known quantities, like temperature, density and N_e .
- 2) How many beads were used per slip-spring? How was that value chosen or determined?
- 3) there should be at least two frictions in the model: between the bead and background, and a friction associated with the hopping rate of the slip-spring. That latter does not appear to be specified anywhere. How do the results depend on these values? How were they chosen?
- 4) Do the slip-springs hop or slide? The latter is problematic from a thermodynamic point of view.
- 5) How do they calculate stress? Do the virtual springs contribute? If so, then do they violate stress-optic rule? If not, then the model presumably violates thermodynamics.
- 6) How do they calculate the dielectric relaxation?

Author's Response to Decision Letter for (RSOS-191046.R1)

See Appendix B.

RSOS-191046.R2 (Revision)

Review form: Reviewer 3

Is the manuscript scientifically sound in its present form?

Yes

Are the interpretations and conclusions justified by the results?

Yes

Is the language acceptable?

Yes

Do you have any ethical concerns with this paper?

No

Have you any concerns about statistical analyses in this paper?

No

Recommendation?

Accept as is

Comments to the Author(s)

The authors have largely answered my questions. I do not agree completely with the approach, but they have now given sufficient details to make the results reproducible, and what the conclusions are based on.

Decision letter (RSOS-191046.R2)

10-Feb-2020

Dear Dr Ma,

It is a pleasure to accept your manuscript entitled "Slip-spring simulations of different constraint release environments for linear polymer chains" in its current form for publication in Royal Society Open Science. The comments of the reviewer(s) who reviewed your manuscript are included at the foot of this letter.

on behalf of Professor Hazel Assender (Associate Editor) and R. Kerry Rowe (Subject Editor)
openscience@royalsociety.org

Reviewer comments to Author:

Reviewer: 3
Comments to the Author(s)

The authors have largely answered my questions. I do not agree completely with the approach, but they have now given sufficient details to make the results reproducible, and what the conclusions are based on.

Appendix A

Dear Prof. Alice Power, Prof. Hazel Assender, Prof. R. Kerry Rowe and dear reviewers:

Thanks for your comments and suggestions for our manuscript entitled with " Slip-spring simulations of different constraint release environments for linear polymer chains" (ID: RSOS-191046). The manuscript has been carefully revised according to your comments. Any details of the revised portions highlighted in red are kept in the marked copy of the revised manuscript. We appreciate Editors/Reviewers' warm work earnestly, and hope that the corrections will meet your expectation.

We are looking forward to your acceptance of this paper to be published in *Royal Society Open Science* . Responses to the reviewers' comments are presented as follows:

Responses to the reviewers' comments

Reviewer 1

Question 1: Table 3: I guess these parameters correspond to 21K in Fig. 9, but the authors should make this clear in the manuscript. Furthermore, "eqn (3)" does not exist, probably (4.1) is meant?

Answer: Yes. These parameters correspond to 21K in Fig.9 and the table head has been changed to "*Fitted parameters used in lifetime version SS_p for PI21K*". The statements of " eqn(3) in this work " were corrected as "**eqn (4.2) in this work**" . We are sorry for that these parts were not clear in the original manuscript.

Question 2: Fig. 7, 8: How did the authors determine the lifetime distribution? How are they able to numerically calculate this quantity with an accuracy better than 10^{-7} ?

Answer: Fig. 7,8 give the $f_d(t)$ curve, from which we divide the relaxation spectrum into power-law region and a single exponential region. Thus we assume the normalized lifetime distribution probability has the following form:

$$p^{CR}(\tau) = \frac{(1-g)\alpha}{\tau_{max}^\alpha - \tau_0^\alpha} \tau^{\alpha-1} H(\tau - \tau_0) H(\tau_{max} - \tau) + g\delta(\tau_d - \tau)$$

The form can be divided into two parts $\frac{(1-g)\alpha}{\tau_{max}^\alpha - \tau_0^\alpha} \tau^{\alpha-1} H(\tau - \tau_0) H(\tau_{max} - \tau)$ and $g\delta(\tau_d - \tau)$, which correspond to CLF mechanism and reptation mechanism respectively.

The parameters $\alpha, g, \tau_0, \tau_{max}, \tau_d$ are constants and should be obtained by fitting the $f_d(t)$

curve using the function form below.

$$f_d(t) = \frac{\int_0^\infty \left[\frac{(1-g)\alpha}{\tau_{max}^\alpha - \tau_0^\alpha} \tau^{\alpha-1} H(\tau - \tau_0) H(\tau_{max} - \tau) H(\tau - t) + \frac{g}{\tau} \delta(\tau_d - \tau) e^{-\frac{t}{\tau}} \right] d\tau}{\int_0^\infty \left[\frac{(1-g)\alpha}{\tau_{max}^\alpha - \tau_0^\alpha} \tau^{\alpha-1} H(\tau - \tau_0) H(\tau_{max} - \tau) + \frac{g}{\tau} \delta(\tau_d - \tau) \right] d\tau}$$

We first simulate 100 PI21K chains for 1e5 timesteps using the probe version SSp, in which the CR mechanism has been turned off. During the dynamics, each entanglement's lifetime from its birth to its disentanglement was recorded and the normalized curve was shown in Fig. 7 and 8, then we fit the curve using the expression above by nonlinear least-squares method.

To make this procedure clear, we rewrite this part as below.

In Page 8, the first line of Section 4, we add

"The lifetime of entanglements is related to the dynamics of polymer chains and one entanglement disappears by either sliding away from the end of the chain which one it is entangled with or vice versa. The former is reptation/CLF and the latter is constraint release(CR). In the original single-chain slip-spring model when one slip-spring slip away from the chain end, there should be one corresponding slip-spring disappearing. For the lifetime version SSp, the lifetime distribution of entanglements should be obtained and the entanglements destructed by reptation/CLF should be compared with CR. "

In Page 9 we add *"We first simulate 100 PI21K chains for 1e5 timesteps using the probe version SSp. During the dynamics, each entanglement's lifetime from its birth to its disentanglement was recorded and the normalized curve fdt was obtained. Using the curve we infer the probability form of lifetime distribution and then fit the curve using one expression discussed below by nonlinear least-squares method. The parameters in the probability expression will be eventually obtained."* after *" Here we use results of sample of PI21K because it is the most time-saving of all and other samples show similar results.*

" We have revised the contents of this part in the revised manuscript.

Because the sampling points were based on entanglements of 100 chains for 1e5 timesteps, which is a large number and 1e-7 occurred during normalization by it. In fact we cannot achieve the accuracy better than 1e-7.

Question 3: page 11: "simulations are in quantitative agreement ..." seems an overstatement, since the loss plateau for 60% (green) in Fig. 11 is about a factor of 2 off.

Answer: Sorry. The statement was indeed overstated. So the original sentence " *Fig.11 gives the simulation results of both monodisperse and bidisperse melts and it can be seen that the results from simulations are in quantitative agreement with experimental measurements from ref .41.* " has been changed to " *Fig.11 gives the simulation results of both monodisperse and bidisperse melts and the results from simulations **show agreement with** experimental measurements from ref .41.* " And according to the other reviewer's suggestion, we have iterated the procedure to find more accurate parameters to fit the experimental results. The new results have replaced the old data in the manuscript.

Question 4: Table 4: Is the exponent alpha really that sensitive that -0.603 and -0.599 give different results from -0.6?

Answer: The exponent alpha is not sensitive in the case in our manuscript. We changed alpha from -0.603 and -0.599 to -0.6 for PS177&60 and the results seem no difference. The results 0.603 and -0.599 were obtained by using nonlinear least-squares method and we left them unchanged. The figure below shows the component relaxation modulus of PS60&177%40 and PS60&177%60 using different alpha values, from which we cannot discriminate one from another.

Fig.S1 The component relaxation function of PS60&177%40 and PS60&177%60 using $\alpha=0.603$ (solid lines) and $\alpha=0.6$ (dashed lines).

We also show the probability density function of $P = \frac{\alpha t^{\alpha-1}}{t_{max}^{\alpha} - t_0^{\alpha}}$ for different alpha in the figure below, which gives almost same distribution.

Fig.S2 The probability density function of $P = \frac{\alpha t^{\alpha-1}}{t_{max}^{\alpha} - t_0^{\alpha}}$ for different alpha using $\tau_0 = 0.45$

and $\tau_{max} = 6000$.

Question 5: A number of typos and awkward wordings should be corrected, such as "mels" -> melts, "simpleness" -> simplicity, etc.

Answer: We are sorry that we have made such mistakes in our manuscript. The whole document has been checked carefully. Thank you for your careful reading and illuminating comments.

Reviewer 2

Question 1: On the topic of orientational cross correlations (OCC) I feel the authors have a misunderstanding of the issue, throughout the paper. OCC in the (multi-chain) Masubuchi work, and in the single chain "slip spring" simulations are quite different things, and cannot be compared one against the other. In the Masubuchi work, the OCC is between different stress-carrying chains in the simulation and arise as a result of force balance at entanglement points between those chains. Since those simulation chains are supposed to represent real chains in a polymer melt, the question is whether real chains also have the orientational cross correlations, and that is a meaningful question (but not one that is answered by the work of the present paper).

On the other hand, the present simulations are all "single chain" simulations in that one chain does not directly exchange forces with others - there can be no OCC between different chains. The OCC are between chains and the virtual springs which hold the chains. The virtual springs are not considered to "really" carry stress. However, the OCC must be included in the Green-Kubo formula to calculate $G(t)$, even though it is considered that the real stress is only carried by the chains. So, this is an entirely different situation from the Masubuchi simulations, and not comparable at all.

Answer: We are sorry that we have done such an improper comparison of OCC with Masubuchi's work from this viewpoint. At first we thought that the chains and virtual springs carry stress and the cross-correlations have similarity with orientational cross

correlations of PCN and we have known the cross-correlations are important in the stress relaxation. The reviewer's comments are lucid and we have removed the sentences related to the comparison with OCC of PCN. The related contents have also been revised in the revised manuscript.

"In contrast to Masubuchi et al.[25], the simulated viscoelastic relaxation intensity is in good agreement with the experiment, which seems no OCC flaws exists in the SSp model.

"in introduction in Page 3 has been updated as

"The simulated viscoelastic relaxation intensity is in good agreement with the experimental data. "

"It is meaningful to compare the results of PCN[25] and SSp simulations. We note that the relaxation time and intensity of viscoelastic relaxation for 43K obtained through PCN are not consistent with the experiment. The simulations in this work don't show this problem but fail to acquire the viscoelastic relaxation time for 21K precisely. We attribute this to that the PCN model is much more coarse-grained than SSp model partially. In the experiment by Matsumiya et al.[22] and simulations in this work, the probed viscoelastic loss is larger than that of the monodisperse melts around the peak whereas PCN simulations show a contrary tendency. Meanwhile, the intensity is molecular weight-dependent from PCN, which is wrong and the SSp results recover this irrelevance found in the experiment. The failure of PCN in predicting the viscoelastic intensity of probe chains is because of the flaws for OCC in the PCN model. So we show the viscoelastic relaxation function with the autocorrelation and cross-correlation contributions in Fig.5 for both probe rheology and monodisperse melts of PI43K (and other samples come to the same conclusion). The cross-correlation contribution of probe melt is similar to that of monodisperse melt in contrast to Fig.4 in ref 25, which validate the conclusion by Masubuchi et al. for the inconsistency of the intensity. "

in Page 7 has been changed to

"For the inaccurate viscoelastic relaxation time for 21K, we check the relaxation time and intensity of viscoelastic relaxation for 43K. We note that the relaxation time and intensity of viscoelastic relaxation for 43K obtained through PCN are not consistent with the experiment. In the experiment by Matsumiya et al.[22] and simulations in this work, the

probed viscoelastic loss is larger than that of the monodisperse melts around the peak whereas PCN simulations show a contrary tendency. Meanwhile, the intensity is molecular weight-dependent from PCN, which is wrong and the SSp results recover this irrelevance found in the experiment. The viscoelastic relaxation function with the autocorrelation and cross-correlation contributions are shown in Fig.5 for both probe rheology and monodisperse melts of PI43K. As noted by the authors, the failure of PCN in predicting the viscoelastic intensity of probe chains is because of the flaws for OCC in the PCN model. Here we only show the results of SSp can reproduce the experimental data and it should not be confused with the OCC in the PCN model. The OCC in the PCN model is a multi-chain property and results from force balance at entanglement points. "

" The conclusions made by Matsumiya et al. [22] have been validated using simulations and **the difference of cross-correlation contributions to stress relaxation from PCN models have also been checked.** "

in Page 8, the conclusion of Section 3 has been changed to

" The conclusions made by Matsumiya et al. [22] have been validated using simulations. **The CR accelerating effect and the coincidence of dielectric and viscoelastic relaxation have been verified using SSp model.** "

"In contrast to PCN results, the cross correlation function has been obtained correctly in both probe rheology and monodisperse melts and thus the inconsistency of viscoelastic relaxation intensity in ref. 25 was avoided. However, the CLF of long chains has been ignored in **both PCN and** SSp simulations, which may affect the results."

in conclusion in Page 12 have been changed to
" **The simulated viscoelastic relaxation intensity is in good agreement with the experimental data of both probe rheology and monodisperse melts.** However, the CLF of long chains has been ignored in SSp simulations, which may affect the results."

Question 2: The list of parameters in Table 1 should include Ne_{ss}.

Answer: We have no Ne_{ss} in our manuscript. The Numbers of beads(N) and entanglements(Z) in our manuscript have been shown in Table2. And Ne_{ss}≈4, which is a typical value in the literature. For clarity, we add "The average number of Kuhn segments

between slip-springs, $N_e = N/Z \approx 4$, which is a typical value in the literature” in Page 4 of Section 2.

Question 3: The version of Reptate referred to is no longer available, and the web address no longer works. I believe there is a new Reptate available.

Answer: Sorry, we have noticed this mistake and new Reptate available website <https://github.com/jorge-ramirez-upm/RepTate> has been inserted at proper position instead of the old one in the manuscript.

Question 4: Page 6: The authors write "It's fair to note that the simulated dielectric relaxation time of probe chains is underestimated systematically". Potentially the problem is the other way around... the authors should fix the timescale of simulations using the "simpler" probe data (where CR is suppressed) and then model the more complicated CR environment of pure melt. Perhaps the pure melt predictions are actually the things that are wrong?

Answer: Yes, the reviewer's comment inspired us a lot. For consistency with the experimental results, we prefer the time scale using in the original manuscript and we hope that won't affect much. The statement of "*It's fair to note that the simulated dielectric relaxation time of probe chains is underestimated systematically. This discrepancy may be related to the chosen MC strategy. However, the MC strategy in this paper doesn't change the conclusions. Due to the chosen MC strategy, the dielectric relaxation time of probe chains and monodisperse chains have almost no difference for P1179K. With increasing the molecular weight of the chains, the CR effect becomes small gradually and we can infer this effect of CR can be neglected when the molecular weight of chains is large enough.*" in Page 6 were corrected as

"For better fit of monodisperse results, the timescale of simulations was chosen as $\tau = 10\mu s$. A negative influence of that is the simulated dielectric relaxation time of probe chains is underestimated systematically. We may chose different MC strategy and tune the timescale to improve the discrepancy. However, the MC strategy in this paper doesn't change the conclusions. The dielectric relaxation time of probe chains and monodisperse

chains have almost no difference for PII79K. With increasing the molecular weight of the chains, the CR effect becomes small gradually and we can infer this effect of CR can be neglected when the molecular weight of chains is large enough. " in Page 6.

Question 5: Page 8, the authors write, "The concrete expression of $f(t)$ can be found in ref. 31.". I think the present paper should be self contained and all necessary equations written here.

Answer: Yes. The concrete expression of $f(t)$ is $f(t) = \int_0^\infty p^{CR} \exp\left(-\frac{t}{\tau^{CR}}\right) d\tau^{CR}$ and we have already revise the part in our manuscript.

"The concrete expression of $f(t)$ can be found in ref. 31." in Page 8 has been changed to "*The concrete expression of $f(t)$ is $f(t) = \int_0^\infty p^{CR} \exp\left(-\frac{t}{\tau^{CR}}\right) d\tau^{CR}$, which means the survival time of entanglements from any moment.* " in Page 8.

Question 6: Page 9 onwards discusses the exponential form for CR events used by Shivokhin. It is worth noting that the exponential distribution was used as a test "theoretical" case to explore the effects of a single dominant CR timescale, without necessarily expecting it to be an accurate model for real chains. Furthermore (figure 9) the exponential distribution is bound to be worst for the shortest chains with fewest entanglements, since such chains have a greater effect of contour length fluctuation. Longer chains should show less discrepancy.

Answer: Yes, Shivokhin et al. proposed the "toy models" to test their theoretical cases and here we cite the exponential form for CR events used by Shivokhin is just to show the results as a comparison when only considering the lifetime distribution of the reptation mechanism. According to the comments of the reviewer, we add "*For short chains, the results using the exponential form distribution in which only the reptation mechanism is considered show big discrepancy and it may be better for longer chains with more entanglements.* " in Page 9.

Question 7: For the binary blend case (figure 11) it would be strongly advisable to iterate the procedure for finding $f(t)$, not just take the first iteration. The presence of short chains can greatly accelerate the relaxation of longer chains, so the entanglement lifetime distribution is quite different from the pure melts of the two components in some (but not all) cases.

Answer: Yes, thanks for your comments. We have tried to iterate the procedure for blend to find a reasonable entanglement lifetime distribution expressing the short chains accelerating the relaxation of longer chains. After many iterations manually, the parameters we obtained are shown in below table. The parameters are fitted by PS60&177%60 simulations.

Table S1. Parameters for PS177&60 melts.

Parameters	PS177	PS60
g	0.849	0.65
α	-0.601	-0.602
τ_0	0.406	0.45
τ_{max}	5300	4800
τ_d	46000	14800

The new viscoelastic storage and loss modulus of bidisperse melts results replace the old curves in Fig. 11.

Figure S3. Simulated viscoelastic storage(broken lines) and loss modulus(solid lines) for

PS60(red),PS177(magenta),PS60&177%40(blue) and PS60&177%60(green). The symbols are experiment data reproduced from ref. 41.

The sentence *As noted by Khaliullin et al.[31], the parameters can be obtained by iteration and here we use the first fitting results at the cost of partial loss of accuracy.* “

in Page 9 has been changed to

"As noted by Khaliullin et al.[31], the parameters can be obtained by iteration and here we use the first fitting results for monodisperse melts. Due to blending, the relaxation environment has changed greatly and we iterate the procedure for bidisperse melts. "

"The fitted parameters of lifetime distribution used in lifetime version SSp for PS177&60 are obtained by simulating PS177 and PS60 monodisperse melts and are listed in Table 4.

"

in Page 11 has been changed to

"The fitted parameters of lifetime distribution used in lifetime version SSp for PS177&60 are obtained by PS60&177%60 simulations iteratively manually and are listed in Table 4. For the cost of computation, we also use the parameters for PS60&177%40. "

Question 8: The magenta line in figure 11 shows a spurious relaxation near $\omega = 10s^{-1}$. What is the source of this?

Answer: We are sorry that it seems a fitting error. We repeat the fitting process many times and the new fitting results are shown below with also the raw simulation results (Schwarzl $G' G''$ from Reptate). The corresponding curves in Fig. 11 in original manuscript have updated using the new fitting results.

Figure S4 Fitted viscoelastic storage(broken lines) and loss modulus(solid lines) for PS177(magenta). The magenta symbols are experiment data reproduced from ref. 41 and green symbols are raw simulation results (Schwarzl G' G'' from Reptate).

Question 9: Page 11, the following two sentences need to be either removed or elaborated on (how would one do this?)

" For example, it may be extended to study the partially dilated tube diameters to check partially DTD."

"Further refinement is necessary and the effects of the parameters of the lifetime distributions should be well understand to achieve more complex CR environment such as nanocomposites with rod-like filler"

Answer: We have removed the first sentences. We think they are interesting directions and we have not thought them over yet.

The sentences of "*Further refinement is necessary and the effects of the parameters of the lifetime distributions should be well understand to achieve more complex CR environment such as nanocomposites with rod-like filler*" are changed to "*Further refinement is necessary and the effects of the parameters of the lifetime distributions should be well understand to achieve more complex CR environment.*"

"It may have applicability to complex materials such as nanocomposites and be used as an efficient verification tool for some theory model such as DTD and partially DTD." has also been removed from the conclusion section.

Response to Associate Editor's comments (Professor Hazel Assender):

We thank for the precious comments from the reviewers and the editor. We have revise the manuscript according to comments by the reviewers. We believe the work in this manuscript has two major innovative points. First, we use single slip-spring model to check the experiment work [Matsumiya et al., *Macromolecules*, 2013, 46, 6067] and we haven't seen SSp model to check the CR contribution to the dielectric relaxation of PI before. The CR mechanism accelerates dielectric and viscoelastic relaxation in monodisperse PI and the effects on viscoelastic relaxation are more pronounced. The coincidence for end-to-end relaxation and the viscoelastic relaxation has also been checked using probe version SSp model. Second, we present a lifetime slip-spring model to simulate dynamics of monodisperse and bidisperse melts. Though the idea of entanglement lifetime is based on the previously published discrete slip-link model(DSM), the single slip-spring model is quite different from DSM model. DSM model is based on a differential Chapman–Kolmogorov equation and is a more coarse-grained one. SSp model use Brownian dynamics simulations to solve stochastic equations of motion numerically. They both are popular models using for the rheology of polymer melts but different from each other. There are many improved models or variant of SSp such as TIEPOS[Abelardo Ramírez-Hernández et al. *Soft Matter*, 2013, 9, 2030–2036] and Multi-chain slip-spring model[Takashi Uneyama et al. *The Journal of Chemical Physics*, 2012,137, 154902]. We borrow the concept of lifetime of entanglements and it's a natural way to fit the lifetime distribution by dynamics without CR first. We show that the lifetime model can be fitted to reasonably reproduce experimental results on monodisperse and bidisperse melts. The model is more flexible and more time-saving than original SSp using pair entanglements. So we believe our work is not a simple derivative of the previous work. We appreciate Editors/Reviewers' warm work earnestly again and we are looking forward to your

acceptance of this paper to be published in *Royal Society Open Science*.

Appendix B

Dear Editor,

Thank you very much for your reply and help. Thanks a lot for the reviewers' comments and their kind suggestions of our manuscript (ID RSOS-191046.R1) entitled "**Slip-spring simulations of different constraint release environments for linear polymer chains**". We provide this cover letter to explain, point by point, the details of our revisions in the manuscript and our responses to the reviewers' comments as follows. In order to make the changes easily viewable for you and the reviewers, in the revised paper, we marked the revision with red color. We appreciate Editors/Reviewers' warm work earnestly, and hope that the corrections will meet your expectation.

We appreciate Editors/Reviewers' warm work earnestly again and we are looking forward to your acceptance of this paper to be published in Royal Society Open Science.

Kind Regards,

Teng Ma

Responds to the reviewer's comments:

Reviewer 1

Comment 1: I am happy with the authors response.

Response:

Thank you for your review very much.

Reviewer: 3

Comments to the Author(s)

The results look as expected, so they probably calculated everything correctly. However, there are not sufficient details given. Also, they have not cited our prior work in the area, but appear to have done significantly less. The idea of modifying the slip-spring simulations in this way is actually taken over from our ongoing work, and it seems that proper credit is missing. Moreover, we have looked at this question before, using our slip link model [Pilyugina, E.; Schieber, J. D. & Andreev, M. Dielectric relaxation as an independent examination of relaxation mechanisms in entangled polymers using the discrete slip-link model *Macromolecules*, 2012, 45, 5728-5743]. There are other papers from our work that are relevant and should also be cited. We have analytic expression for $f_d(t)$ which might be compared to those found here, for example. We also have a more fine-grained version of our model that should be very similar.

In principle, I like that they are working to make the slip-spring simulations more rigorous, but am disappointed that they are largely ignoring our contributions that have largely solved all of these problems. Have they found any new conclusions? Finally, I have a number of important technical issues.

Response:

Thanks for your professional comments. It's our great honor to receive your review and instructions. Indeed, we got a lot of help in getting started with rheology of entangled melts from your papers.

According to your suggestion, we add details of the slip-spring model used in our paper which is highlighted in red in the revised manuscript. we rewrite this part as below.

In Page 3, the six line of Section 2, we add

“The potential of the Rouse chains is : $U = \frac{3k_B T}{2b^2} \sum_{i=0}^{N-1} (\mathbf{r}_{i+1} - \mathbf{r}_i)^2$ and the potential of the virtual springs is : $U_{ss} = \frac{3k_B T}{2N_s b^2} \sum_{j=1}^Z (\mathbf{a}_j - \mathbf{r}_{S_j})^2$, in which b is the monomer size and N_s is the stiffness of virtual springs. ”

In Page 3, we add an additional paragraph to describe the details of our implementation:

“We adopt the discrete version of slip-spring model as described in ref. [31,37] and use following steps to complete the simulations in this article:

1. Sample the model: we generate the polymer conformations by sampling from the Gaussian distribution using $P_{\text{eq}}(\{\mathbf{r}_i\}) = \left(\frac{3}{2\pi b^2}\right)^{3(N-1)/2} \exp\left[-\sum_{i=1}^{N-1} \frac{3}{2b^2} (\mathbf{r}_{i+1} - \mathbf{r}_i)^2\right]$, generate Z slip-springs by uniform distribution and generate each anchoring point from the Gaussian distribution using $P_{\text{eq}}(a_j|\{r_i\}, S_j) \equiv \left(\frac{3}{2\pi N_s b^2}\right)^{3/2} \exp\left[-\frac{3}{2N_s b^2} (\mathbf{r}_{S_j} - \mathbf{a}_j)^2\right]$.

2. Update the conformation of polymers via Brownian dynamics using:

$\xi \mathbf{r}_i(t + \Delta t) = \left[\frac{3k_B T}{b^2} (\mathbf{r}_{i+1} - 2\mathbf{r}_i + \mathbf{r}_{i-1}) + \mathbf{f}_i(t) + \frac{3k_B T}{N_s b^2} \sum_{j: S_j=i} (\mathbf{a}_j - \mathbf{r}_i)\right] \Delta t$, in which ξ is the friction coefficient of monomer and $\mathbf{f}_i(t)$ is Gaussian white noise with zero mean, variance

$$\langle f_i(t) f_j(t') \rangle = 2k_B T \xi I \delta(t - t') \delta_{ij}.$$

3. Update the configuration of the slip-springs via Monte Carlo moves:

A discrete slip-spring jump is attempted on average per slip-spring of each chain at each time step and one bead can only occupy one single slip-spring. For this hopping rate, the friction of the slip-springs is assumed to be $\xi_{ss} = \frac{k_B T}{b^2} \tau_{ss}$ here, in which τ_{ss} is equal to the time step dt

here. A hopping move of the j^{th} slip-spring from \mathbf{r}_{S_j} to a neighbor one $\mathbf{r}_{S_{j\pm 1}}$ is accepted with probability $P_{\text{accept}} = \min(1, \exp(-\Delta U))$. The probability to propose the move $S_j \rightarrow S_{j+1}$ or $S_{j+1} \rightarrow S_j$ is equal to $\frac{1}{2}$. If one slip-spring reached the end of the chain and it gets rid of the chain with probability $\frac{1}{2}$. We hold constant entanglement number during the simulations and how to recreate the new slip-spring depends on the constraint environment which will be described in detail below.”

We are sorry that we haven't cite your prior works reasonably before and we didn't intend to offend or to ignore these works deliberately. We have read carefully the relevant papers (listed as following) again and cite them in the appropriate places in the text.

In the second paragraph of the introduction section we add:

“One pioneering simulation work on checking the dielectric relaxation mechanism in entangled polymers is achieved by Pilyugina et.al.[22] They used Detailed Slip-link Model (DSM) which is a well-defined mathematical object and has been used and extended widely in the literature[23-26]. DSM models have been valid to predict both dielectric and viscoelastic relaxations for linear monodisperse, linear bidisperse and monodisperse star-branched melts. Their results suggest no CR contribution to the end-to-end fluctuation of monodisperse and bidisperse linear PI. The success of DSM model and the like inspired us to find similar approach to separate the sliding dynamics and CR dynamics for slip-spring models. At the same time, the use of life-time distribution probability makes it much more compute-efficient, flexible and adaptable. For its advantages, DSM and the like models can be used as a guide for other models and can be extended to non-linear flow, polydisperse phases, coarse graining, etc.[27-29]”

The following references were added to the “References” section in the revised manuscript.

22. Pilyugina E, Andreev M, Schieber JD. 2012 Dielectric relaxation as an independent examination of relaxation mechanisms in entangled polymers using the discrete slip-link model. *Macromolecules* 45, 5728–5743.

23. Katarova M, Kashyap T, Schieber JD, Venerus DC. 2018 Linear viscoelastic behavior of bidisperse polystyrene blends: experiments and slip-link predictions. *Rheologica Acta* 57, 327–338.

24. Valadez-Pérez N, Taletskiy K, Schieber J, Shivokhin M. 2018 Efficient Determination of Slip-Link Parameters from Broadly Polydisperse Linear Melts. *Polymers* 10, 908.

25. Taletskiy K, Tervoort TA, Schieber JD. 2018 Predictions of the linear rheology of polydisperse, entangled linear polymer melts by using the discrete slip-link model. *Journal of Rheology* 62, 1331–1338.

26. Andreev M, Schieber JD. 2015 Accessible and quantitative entangled polymer rheology predictions, suitable for complex flow calculations. *Macromolecules* 48, 1606–1613.

27. Khaliullin RN, Schieber JD. 2010 Application of the slip-link model to bidisperse systems. *Macromolecules* 43, 6202–6212.

28. Andreev M, Feng H, Yang L, Schieber JD. 2014 Universality and speedup in equilibrium and nonlinear rheology predictions of the fixed slip-link model. *Journal of Rheology* 58, 723–736.

29. Andreev M, Khaliullin RN, Steenbakkens RJ, Schieber JD. 2013 Approximations of the discrete slip-link model and their effect on nonlinear rheology predictions. *Journal of Rheology* 57, 535–557.

We also fit $f_d(t)$ using $f_d(t) := \frac{\dot{f}(t)}{\dot{f}(0)} = \frac{\int_0^\infty \frac{p^{CD}(\tau^{CD})}{\tau^{CD}} \exp\left(-\frac{t}{\tau^{CD}}\right) d\tau^{CD}}{\int_0^\infty \frac{p^{CD}(\tau^{CD})}{\tau^{CD}} d\tau^{CD}}$, which is the analytic

expression in [Khaliullin RN, Schieber JD. 2009 Self-Consistent Modeling of Constraint Release in a Single-Chain Mean-Field Slip-Link Model. *Macromolecules* 42, 7504–7517.]. We got $g = 0.67, \alpha = 0.35, \tau_0/dt = 0.63, \tau_{max}/dt = 2000, \tau_d/dt = 5200$. So far we only use these parameters in our slip-spring model and the relaxation function and deleted entanglement numbers are depicted in figures below. We found the relaxation is slower than the results from slip-spring model and the destruction by CR is far less than that of SD. In the revised manuscript, Fig.9 and Fig.10 are updated.

Fig.S1 Simulated relaxation functions for PI 21K of different versions: probe version, coupling entanglements version and lifetime version. The results using exponential distribution and parameters obtained by fitting eqn (19) in ref. 39 are also shown.

Fig.S2 Simulated numbers of entanglements destroyed for PI 21K with 100 chains. The solid lines represent destruction by reptation/CLF and broken lines represent destructions by CR. The red and magenta curves represent results using parameters fitted by eqn(4.2) in this work and eqn(19) in ref. 39, respectively. The blue curves represent results using lifetime from exponential distribution.

We add following description in Paragraph 1 on Page 11:

“We also obtain the parameters by fitting eqn (19) in ref. [39]. The fitted parameters are listed in Table 3 and the relaxation function and deleted entanglement numbers are depicted in Fig.9 and Fig.10. In comparison to coupling entanglements version results, the relaxation is slow and destruction by CR is underestimated. The difference may be from the different description of dynamics between the SSp model and the DSM model. The comparison between these two models may be an interesting topic in the future research.”

We are working to improving the single slip-spring model and we are very sorry again for not fully citing your great contributions to these issues. As mentioned before, your works guide us into the door to rheology of entangled melts and we cherish your contribution.

Question 1 : Why are the authors treating the modulus as an adjustable parameter? It should be determined completely from other known quantities, like temperature, density and N_e .

Answer : We treat the modulus as an adjustable parameter for mapping the stress of the model to those of the experimental data. In fact, the model used in our manuscript is based on [Shivokhin ME, Read DJ, Kouloumasis D, Kocen R, Zhuge F, Bailly C, Hadjichristidis N, Likhtman AE. 2017 Understanding Effect of Constraint Release Environment on End-to-End Vector Relaxation of Linear Polymer Chains. *Macromolecules* 50, 4501–4523.] and we treat the

modulus as a fitted value following their process. Based on your comment, we also determine the modulus using $G_0 = \frac{\rho RT}{M_0}$, with ρ , R , T , M_0 being density, universal gas constant, absolute temperature and the molecular weight represented by one bead, respectively. At present, we use $\rho = 920 \text{ kg/m}^3$, $T = 313\text{K}$, $R = 8.314 \text{ m}^3 \text{ PaK}^{-1} \text{ mol}^{-1}$ and $M_0 = 1 \text{ kg/mol}$ for PI. The computed $G_0 = 2.39 \text{ MPa}$, which is too smaller than the fitted 3.3MPa we used in the manuscript. To solve this problem, we may need to adjust the parameter M_0 and use a different Monte Carlo algorithm for the motion of the slip-spring and we hope we can adopt this approach in our further research. Thanks for your comment very much.

To make it clear, we add following part in the first paragraph in the “Results” section:

As proposed by the reviewers, the modulus should be determined using $G_0 = \frac{\rho RT}{M_0}$ with values $\rho = 920 \text{ kg/m}^3$, $T = 313\text{K}$, $R = 8.314 \text{ m}^3 \text{ PaK}^{-1} \text{ mol}^{-1}$ and $M_0 = 1 \text{ kg/mol}$ for PI used here. The computed $G_0 = 2.39 \text{ MPa}$, which is smaller than the fitted 3.3MPa and here we use 3.3MPa for better mapping to the experimental data.

Question 2 : How many beads were used per slip-spring? How was that value chosen or determined?

Answer : Approximately four beads were used per slip-spring. This parameter was represented by $N_e = N/Z$, where N is the beads number of Rouse chains and Z is the number of slip-springs. This value was originally proposed by Likhtman in his paper [Macromolecules 2005, 38, 6128-6139] and we found it is widely used in the literature and shows good agreement with experiments. We have slightly changed the N_e for different system for the best fitting purpose. We have also simulated all system with $N_e = 4$ and the results (the results were not shown in the paper) are close, which we think doesn't change the conclusions made in our paper. This flexibility in parameter selection is not uncommon in this field using slip-spring like models, for example the work of [Soft Matter, 2013, 9, 2030–2036].

We make it clear in the revised manuscript as below:

" The average number of Kuhn segments between slip-springs, $N_e = N/Z \approx 4$, which is a typical value used in the literature. The slightly change of N_e here is made to better match the experimental results".

Question 3: there should be at least two frictions in the model: between the bead and background, and a friction associated with the hopping rate of the slip-spring. That latter does not appear to be specified anywhere. How do the results depend on these values? How were they chosen?

Answer : Yes, there should be two frictions in the model, which makes more sense. So far, we are sorry that we have only one friction associated with monomer beads explicitly. We conduct a discrete slip-spring jump attempt on average per slip-spring of each chain at each time step and thus the friction associated with slip-springs is hidden. This strategy is used in [Macromolecules 2017, 50, 4501–4523]. From this hopping rate, the slip-spring friction $\gamma_{ss} \sim \frac{k_B T}{b^2} \tau_{SS}$ is assumed to be 0.05 used in our manuscript. We are sorry that we haven't make it clear in the manuscript and many thanks for your comments. We realized there exist advanced models such as the multi-chain polymer slip-spring model with fluctuating number of entanglements [THE JOURNAL OF CHEMICAL PHYSICS 143, 243147 (2015), THE JOURNAL OF CHEMICAL PHYSICS 146, 014903 (2017)] , conducting Monte Carlo (MC) moves every τ_{SS} time units. τ_{SS} is associated with slip-spring friction coefficient γ_{ss} . The model learned from some ideas of DSM and we found the time scale τ_{SS} need to shift one Kuhn step through an entanglement is well-defined in DSM. we hope we can discuss the effect of slip-spring friction coefficient γ_{ss} on the results in our further research. The corresponding part have been revised in the manuscript as below:

Question 4: Do the slip-springs hop or slide? The latter is problematic from a thermodynamic point of view.

Answer : The slip-springs hop using Metropolis Monte Carlo moves as described in [Macromolecules 2017, 50, 4501–4523] and [Likhtman, A. E. Viscoelasticity and Molecular Rheology. In A Comprehensive Reference; Elsevier B.V.: Amsterdam, 2012; pp 133–179.].

In the revised manuscript, we make it clear in Section 2. Here we transcribe the corresponding part:

A discrete slip-spring jump is attempted on average per slip-spring of each chain at each time step and one bead can only occupy one single slip-spring. A hopping move of the j^{th} slip-spring from \mathbf{r}_{S_j} to a neighbor one $\mathbf{r}_{S_{j\pm 1}}$ is accepted with probability $P_{\text{accept}} = \min(1, \exp(-\Delta U))$. The probability to propose the move $S_j \rightarrow S_{j+1}$ or $S_{j+1} \rightarrow S_j$ is equal to $\frac{1}{2}$.

Question 5: How do they calculate stress? Do the virtual springs contribute? If so, then do they violate stress-optic rule? If not, then the model presumably violates thermodynamics.

Answer : We compute the modulus using

$$G(t) = \frac{V}{k_B T} \frac{1}{3} \left\langle \sum_{\alpha=1}^2 \sum_{\beta>\alpha}^3 \sigma_{\alpha\beta}^R(t) (\sigma_{\alpha\beta}^R(0) + \sigma_{\alpha\beta}^{sL}(0)) \right\rangle, \text{ and } \sigma_{\alpha\beta}^R, \sigma_{\alpha\beta}^{sL} \text{ are}$$

$$\sigma_{\alpha\beta}^R = - \frac{3k_B T}{Vb^2} \left\langle \sum_{i=1}^N (r_{\alpha,i} - r_{\alpha,i-1}) (r_{\beta,i} - r_{\beta,i-1}) \right\rangle \text{ and}$$

$$\sigma_{\alpha\beta}^{sL} = - \frac{3k_B T}{N_s Vb^2} \left\langle \sum_{j=1}^Z (r_{\alpha,x_j} - a_{\alpha,j}) (r_{\beta,x_j} - a_{\beta,j}) \right\rangle, \text{ respectively. These expresses are widely used}$$

in the literature. As explained in the literature [Ramírez, Jorge, Sathish K. Sukumaran, and Alexei E. Likhtman. "Significance of cross correlations in the stress relaxation of polymer melts." *The Journal of chemical physics* 126.24 (2007): 244904.] and [Likhtman, Alexei E. "Viscoelasticity and molecular rheology." *Polymer Science: A Comprehensive Reference* 1 (2012): 133-179.], we may employ $\sigma_{\alpha\beta}^R(t)$ as the stress tensor of the system. However, the cross-correlations between the stress of virtual springs and the stress of Rouse chains are included for computing $G(t)$ for its un-negligible contributions. We believe such a definition does not violate the stress-optic rule. Although further improvement of the model may be possible, we believe the obtained rheological data were reasonable here.

Question 6: How do they calculate the dielectric relaxation?

Answer : The dielectric relaxation modulus were got from the end-to-end vector autocorrelation functions , i.e. $\varepsilon(t) = \frac{\langle \mathbf{P}(t)\mathbf{P}(0) \rangle}{\langle P^2(0) \rangle}$. The cross-correlation term is not considered for relaxation of end-to-end vector. We are sorry for that it is not clear in the previous manuscript.

We add this part in the methods section as below:

"The stress relaxation function $G(t)$ is computed using the correlation functions of shear stress, including autocorrelation contributions and cross-correlations contributions from the slip-springs. The dielectric relaxation function $\varepsilon(t)$ is computed using the end-to-end vector autocorrelation functions for each chain. The expression of $G(t)$, $\varepsilon(t)$, stress related to Rouse potential $\sigma_{\alpha\beta}^R(t)$ and slip-spring $\sigma_{\alpha\beta}^{SL}(t)$ are given below as ref[42,48]:

$$G(t) = \frac{V}{k_B T} \frac{1}{3} \left\langle \sum_{\alpha=1}^2 \sum_{\beta>\alpha}^3 \sigma_{\alpha\beta}^R(t) (\sigma_{\alpha\beta}^R(0) + \sigma_{\alpha\beta}^{SL}(0)) \right\rangle$$

$$\varepsilon(t) = \frac{\langle \mathbf{P}(t)\mathbf{P}(0) \rangle}{\langle P^2(0) \rangle}, \text{ in which } \mathbf{P} \text{ is the end-to-end vector of Rouse chains.}$$

$$\sigma_{\alpha\beta}^R = - \frac{3k_B T}{Vb^2} \left\langle \sum_{i=1}^N (r_{\alpha,i} - r_{\alpha,i-1}) (r_{\beta,i} - r_{\beta,i-1}) \right\rangle$$

$$\sigma_{\alpha\beta}^{SL} = - \frac{3k_B T}{N_s V b^2} \left\langle \sum_{j=1}^Z (r_{\alpha,x_j} - a_{\alpha,j}) (r_{\beta,x_j} - a_{\beta,j}) \right\rangle,$$

where V is the volume of the melts and the indices $\{1,2,3\}$, α, β represent the Cartesian coordinates."